# Strategic prioritization of blockchain diffusion factors by an integrated fuzzy Delphi-Dombi LMAW model: A case study in Bangladesh healthcare sectors

Nasif Morshed, Raad Shahamat Labib, Mushfiqur Rahman, Kazi Md. Tanvir Anzum*

Department of Industrial Engineering and Management, Khulna University of Engineering & Technology, Khulna, Bangladesh

* tanvir@iem.kuet.ac.bd

## Abstract

Over the recent years, blockchain technology has been an essential practice in enhancing profitability in healthcare supply chains, especially in emerging economies. Nevertheless, regulatory uncertainty, management resistance and inadequate technological infrastructure are major setbacks. This study evaluates the diffusion factors of blockchain technology in healthcare sector of Bangladesh within a two-stage method; first, combining literature review and expert opinions, 26 significant factors were identified, which were later reduced to 22 through Fuzzy Delphi Method across 5 dimensions (Organizational, Technological, Environmental, Managerial, Individual). In the second stage, the factors ranked in importance using the Dombi Logarithmic Methodology of Additive Weights (LMAW) were consequently subjected to sensitivity analysis to ensure the robustness of results. The results indicate that the most impactful drivers are Top Management Support (0.0501), Organization Readiness (0.0485) and Trust Among Stakeholders (0.0481) for all the healthcare organizations whereas Smart Contract Automation (0.0368) and Innovation Pressure (0.0362) got the least importance and indicate that the sector is in need of strategic leadership and infrastructure rather than advanced innovations. Theoretically, this study advances technology adoption literature by demonstrating that fundamental organizational readiness and leadership support act as non-compensatory prerequisites for blockchain diffusion in emerging economies. Managerially, the proposed framework empowers healthcare executives and policymakers with an actionable roadmap for targeted resource allocation, facilitating strategic blockchain integration to enhance supply chain transparency and resilience.

**Data availability statement:** All relevant data are within the manuscript and its Supporting Information files.

**Funding:** The author(s) received no specific funding for this work.

**Competing interests:** The authors have declared that no competing interests exist.

## Introduction

Blockchain is a noteworthy technology within the context of the discussion about process recoverability or resilience; it may be viewed as the capability of medical systems to withstand disruptions and change and maintain the operation of the system [1]. Remarkably, it is impressive how the application of blockchain technology into the enhancement of the health care supply chain can be implemented. The continuation of blockchain and other industry 4.0 technologies towards improving the resilience of the healthcare supply chain is only at an early phase, and considerable geographic concentrations have been identified although there remains considerable research deficiencies especially on agility, openness, protection and faithfulness in a supply chain [2]. This is consistent with a more general trend to implement sophisticated, intelligent technologies in healthcare, including the application of 3D graph deep learning to intelligent rehabilitation [3]. This smart system integration is particularly applicable to the support of human-machine interaction in such applications as hand function rehabilitation that is increasingly dependent on 3D deep learning and point cloud data [4]. To support better supply chains in healthcare with a less disruption, more flexible supply chains, this is enhanced by creating effective healthcare supply chains, with blockchain technology increasing visibility and expediency in supply chain partnership and effectiveness [5]. The COVID-19 pandemic in the entire world has posed numerous challenges to the health sector. A few of the problems, which are yet to be tackled by healthcare sector, are supply chain breakdowns, inefficiency and environmental problems [6]. Thus, resolving these issues by implementing blockchain technology into the supply chains will benefit Healthcare providers and consumers alike [7]. The blockchain technology is potentially able to change healthcare but its usage is rather low with about 11 percent of healthcare institutes using blockchain solutions per Statista Research Department [8]. The factors affecting successful implementation are one of the main priorities of the up-to-date research [9]. Blockchain technology has enormous potential to improve supply chain efficiency, traceability and transparency in Bangladesh's healthcare industry. Even though these advancements are fundamental, blockchain adoption is happening at a very modest pace. The main challenges include high startup costs, a shortage of staff with the requisite technical skills, and resistance to change. To fully explore its potential and clear the path for an effective, transparent, and sustainable healthcare sector in Bangladesh, more research into the adoption of blockchain in different nations and industries, such as healthcare, telemedicine, and medical research, is crucial to comprehending and resolving these issues. Crucially, there is still not a prioritized, actionable plan for blockchain to spread successfully in developing countries with limited resources like Bangladesh. This means that we need to do more than just list the barriers; we also need to make a clear hierarchy for allocating resources.

The main goal of this study is to fill in a gap in the research by showing how to find the right conditions for blockchain technology to spread successfully into the Bangladeshi healthcare supply chain. In this case, blockchain diffusion means that an organization is ready to adopt or apply blockchain technology in a way that leads to

the desired results, such as more openness, less fraud, and more efficient supply chains [10]. The study focuses especially on Bangladesh's healthcare system. Bangladesh's healthcare sector is vital to the country's health, but it has unique challenges in achieving international standards, optimizing operations, and ensuring supply chain integrity. The purpose of this research is to analyze an environment that supports the adoption of new technologies and also identifies hindrances in terms of blockchain and healthcare supply chains in Bangladesh. Bangladesh is a developing country with a growing healthcare sector, therefore Blockchain integration is considered as a promising but challenging avenue. The primary goal of the research is to undertake an assessment of the Organizational aspects, technological aspects, Environmental factors, Management aspects and individual factor conditions in order to provide recommendations that can assist effective blockchain technology diffusion and address sector-specific concerns. Existing Multi-Criteria Decision-Making (MCDM) approaches, such as Analytic Hierarchy Process (AHP) or Best-Worst Method (BWM), often struggle to effectively capture the vagueness and high uncertainty intrinsic to expert decision-making in emerging economies and may not adequately address the non-linear interdependencies among complex diffusion factors. This research wants to discover certain issues and opportunities concerning blockchain adoption in Bangladesh's healthcare sector by carrying out case studies so as to provide policymakers, regulators, and firms with insights on how the technology may boost productivity levels along all the health product lines through blockchain technology. In addition, the study also intends to provide insights that blockchain technology could be used to make practical policy propositions aimed at boosting creativity in the Bangladeshi healthcare system and maintaining it over time, for example by enhancing adaptability and efficiency of pharmaceutical supply chains.

This study uses Fuzzy Delphi Method (FDM) to capture systematic agreement among the experts regarding the issues that affect adoption of blockchain in the healthcare supply chain. Based on a traditional Delphi methodology, the FDM is an effective, simple-to-iterate method that uses fuzzy logic to collect expert opinions and eliminate the ambiguity that comes with complex multi-criteria choices [11,12]. With the assistance of expert groups, it has been widely used in a variety of sectors to identify and prioritize a small number of criteria, achieving both methodological strength and practical value. In order to ascertain the relative significance of the criteria for the proliferation of blockchain technology into the healthcare supply chain, this study also aims to offer an efficient fuzzy Dombi-based MCDM model that integrates the Logarithmic Method of Additive Weights (LMAW) [13]. The Dombi-based LMAW approach is used to compute the weight coefficients of the criteria with great accuracy and precision [14]. The novel LMAW technique created by Dombi has numerous advantages over traditional techniques. In order to provide a more flexible and precise weighting method, it makes use of the Dombi operator, which can efficiently account for the ambiguity and uncertainty inherent in decision-making processes [15]. Furthermore, by incorporating relationships between decision variables, the model's accuracy and dependability are significantly increased. By enabling greater discriminating across criteria, this approach improves the decision-making framework and raises the overall precision and robustness of the results [16]. Furthermore, the Dombi Operator-based LMAW technique is especially well-suited for complex and dynamic situations due to its capacity to incorporate nonlinear interactions among criteria, which guarantees a more thorough and realistic assessment.

The remainder of the paper is arranged as follows: Section 2 reviews the relevant literature concerning Blockchain in the Healthcare supply chain and the FDM-Dombi model and identifies the significant research gap. Section 3 defines the problem that need to be addressed in this study and identifies the diffusion factors. Section 4 details the methodology framework which explains research strategy including case selection criteria and data collection techniques used. Section 5 presents each of the case studies together with their analysis along with a cross-case assessment aimed at establishing similarities as well as differences among them. Section 6 undertakes a sensitivity analysis where robustness of our findings is reviewed through testing influence by different assumptions and parameters on it. Section 7 presents Discussion with theoretical and managerial implications. Finally, Section 8 concludes with key findings and limitations with recommendations for future research.

## Literature review

### Blockchain in healthcare supply chain

The traditional supply chain of the healthcare industry is being transformed through the use of a blockchain by ensuring data security and decentralization. Blockchain integration in the pharmaceutical supply chain leads to the elimination of counterfeit drugs as the system helps keep track of each product as it moves through the supply system from the manufacturer to the consumer [17]. This unalterable record provides openness, a key element in combating drug counterfeiting. Implementation of blockchain and smart contracts also helps to automate other operations, such as the verification of orders and payments, as well as increasing the level of security of health care activities [18]. However, there are number of limitations or challenges that has come in light while implementing the blockchain in healthcare supply chains. Including a range of stakeholders, a healthcare network is inherently large, and blockchain has scale limitations. The final issue is to overcome these infrastructural and technological barriers to achieve broader implementation [19]. This is especially true for pharmaceuticals where product identity is an important aspect due to potential dangers posed to human life [20]. Furthermore, blockchain has a positive impact on the electronic record of medical products and facilitates the tracking of the goods. Thus, through the implementation of blockchain, the healthcare providers make certain that the counterfeits are very limited since each transaction is recorded and authenticated. The increased transparency, in this case, provides an assurance that patients are administered authentic drugs [21]. The primary obstacles to blockchain implementation in healthcare are the volatility of the data security issue and the insufficient involvement of stakeholders. In promoting the usage of blockchain, such challenges need to be overcome [22]. Moreover, the application of blockchain with the present-day environments, like IoT and cloud-based structures, will present concerns for effective integration [23].

### Fuzzy Delphi and Dombi LMAW Model

Fuzzy Delphi Method (FDM) is an advanced form of consensus building technique which augments traditional version of Delphi and considers a theory of Fuzzy sets so as to represent and measure the uncertainty expressed by experts and linguistic preferences [11]. FDM has been successfully applied in many fields such as education, behavioral science and technology management to identify, confirm, and rank factors in a systematic fashion using repeated expert surveys and fuzzy aggregation of answers [24]. The DOMBI model LMAW was introduced to evaluate Metaverse transformations in logistics using the Logarithmic Methodology of Additive Weights that integrates Dombi norms [25]. Spherical fuzzy sets are better suited for handling uncertainty and can therefore assist groups in making improved choices. The greater application of fuzziness in decision-making is because Dombi-t norm and t-conorm enhance aggregation in complex judgements [26]. In dealing with uncertain linguistic information space for improving accuracy and flexibility in decision-making, there has been an introduction of evolutionary algorithm based approach for constructing fuzzy Archimedean Dombi normed weighted operators [27]. During the decision making process, this technique enables an adequate capturing as well as ranking for criteria interdependences so that green suppliers can get selected with less mistakes [28].

### Research gap

Blockchain-based solutions has been explored in infrastructure resilience and separately there are few studies that looks into the role of innovation in resilience [29,30]. In this regard, the study seeks to overcome this gap by creating and running a LMAW-based model focused on Dombi using the systematic approach of rank factors of blockchain adoption. In addition to providing decision makers with a solid tool for making decisions in the complex world of blockchain technology, this strategy goes beyond a theoretical understanding of the barriers and enablers of blockchain integration.

Most studies currently in existence employ traditional MCDM methods, such as the AHP or BWM, which, although reliable, frequently presume linear correlations and may oversimplify the subjective character of expert opinions, particularly in high-uncertainty settings like the healthcare industry in Bangladesh. This work pioneers the integration of the

FDM with the Dombi LMAW in order to overcome this. The FDM achieves systematic expert consensus under fuzzy logic, thereby reducing the factor list to contextually relevant criteria. The Dombi operator provides more flexibility to model the non-linear interdependencies between factors and effectively accounts for the imprecision and vagueness inherent in linguistic expert evaluations, making the Dombi LMAW method superior for the subsequent prioritization. This ensures a more accurate and context-sensitive weighting system than traditional additive or multiplicative aggregation techniques. In complex fuzzy systems, this methodological combination offers a more reliable and theoretically valid tool for strategic decision-making.

### Problem definition

This case study centers on Bangladesh's healthcare system, which is essential to the country's health. Blockchain technology has a lot of potential to improve healthcare supply chain management's transparency, efficiency, and fraud reduction, but its implementation is being seriously impeded by resource limitations, legacy systems, and organizational opposition unique to poor nations. From a theoretical standpoint based on fuzzy systems, the diffusion factors that have been identified can be considered as crucial elements whose prioritization and selection are necessary to build a solid decision model for strategic success. Therefore, this study views the factor reduction process via FDM as analogous to heterogeneous feature selection under uncertainty [31], and the subsequent weighting via the Dombi LMAW model as an advanced mechanism for feature weighting and prioritization in a complex fuzzy environment. The fundamental problem addressed is thus the MCDM challenge under uncertainty: specifically, how to accurately weight and prioritize a large, competing set of diffusion factors (i.e., critical features) to determine the optimal, resource-efficient strategy for maximizing the blockchain diffusion effect in the Bangladeshi healthcare supply chain. This strategic prioritization is necessary to move from theoretical discussions of barriers to a clear, actionable policy roadmap.

### Identification of blockchain diffusion factors

A comprehensive literature review was conducted to find out the most influential factors that could cause the increase in blockchain diffusion in the healthcare supply chain of Bangladesh. The theoretical background of blockchain diffusion was premised on past empirical research that has not only addressed the blockchain adoption in different fields (such as manufacturing, logistics, and healthcare) but also had a focus on blockchain adoption in the different fields. To a greater extent, the studies by [10,32,33] gave very important information on the contextual, technological and managerial complexities that blockchain implementation faces in emerging economies. Upon the foundation of these and some other relevant research, the existing study has determined five core categories of factors that enhance blockchain diffusion considerably: Organizational Factors (OF), Technological Factors (TF), Environmental Factors (EF), Managerial Factors (MF), and Individual Factors (IF). The final list of contextually relevant factors, along with their detailed descriptions and sources, are consolidated in the streamlined Table 3.

### Organizational factors (OF)

This dimension, encompassing internal capacities and preparedness, covers the following factors: Organizational Readiness (OF1), Implementation Cost (OF2), Resistance to Change (OF3), Top Management Support (OF4), Lack of ROI Clarity (OF5) and Lack of Practical Use Cases (OF6) [33–35].

### Technological factors (TF)

This dimension deals with the blockchain system's fundamental technical features and compatibility problems. These factors encompass: Blockchain Scalability (TF1), Interoperability with Legacy Systems (TF2), Smart Contract Automation (TF3), Integration with IT Infrastructure (TF4), Standardization Issues (TF5), Traceability (TF6) and Cybersecurity Risks (TF7) [10,34–38].

### Environmental factors (EF)

External pressures and regulatory circumstances are related to environmental factors. The identified factors are: Competitive Pressure (EF1), Innovation Pressure (EF2), Vendor Readiness (EF3), Regulatory and Legal Clarity (EF4) and Performance Expectancy (EF5) [34,35,39].

### Managerial factors (MF)

The Managerial Factors focusing on strategic leadership and internal governance, consist of: Transparency and Immutability (MF1), Perceived Benefits (MF2), Trust Among Stakeholders (MF3) and Disintermediation (MF4) [10,40].

### Individual factors (IF)

User-centric and behavioral elements are captured by the following factors: Lack of Awareness (IF1), Trust in Technology (IF2), Social Influence (IF3) and Effort Expectancy (IF4) [33,41].

### Research methodology

A two-phase methodology is used in the study framework to determine and rank the major elements impacting the adoption of blockchain technology in healthcare supply chains. A thorough assessment of the literature and expert consultations are used to first identify important factors. Fuzzy-based scoring is then used to systematically rank these aspects, allowing for a thorough assessment of their relative importance. In order to ensure dependability in the prioritization process, sensitivity analysis is finally performed to assess the stability and robustness of the outcomes. This approach makes it easier to comprehend in detail the key factors influencing blockchain adoption in the healthcare industry. The two-stage research framework is depicted in Fig 1.

### Participants and study design

An observational and cross-sectional analysis were used to conduct the investigation. A panel of ten experts from four distinct healthcare organizations was chosen using a purposive and judgmental sampling technique. These experts were divided into five categories for analysis. This non-probabilistic method was used to make sure that each participant had in-depth knowledge of a particular field that was pertinent to the study questions. The ten participants' familiarity with the issues that arise in the operation of the healthcare industry adds to the data's trustworthiness. To mitigate potential biases, the expert pool was intentionally diversified to cover different organizational types (Government, Private, Academic and Technical) and functional roles, including Clinical (Senior/Junior Doctor, Nurse), Managerial (Hospital Manager, Supply Chain Manager), Technical (IT Specialist, Pharmacist) and Policy (Govt. Health Official, Academic Researcher). Instead of utilizing statistical significance testing, which is typically irrelevant for consensus-building procedures using judgmental sampling in a qualitative-dominant framework, the sample size of 10 experts was chosen based on the expert-driven consensus principle inherent in the Delphi method. In such methodological frameworks, the validity of the results depends on the specialized knowledge of the participants rather than the sample size; as noted by [42], a panel of 10–15 experts is generally sufficient to yield stable results and reach a meaningful consensus in a Delphi-based study. Furthermore, this size was sufficient to cover all four critical functional roles (Clinical, Managerial, Technical, and Policy) across different organizational types (Government, Private, Academic) essential for ensuring construct validity and mitigating bias. Anonymity was maintained and all participants provided written informed consent after receiving a study briefing. Table 1 summarizes participant profiles and their relational assessments. This study involved the prospective recruitment of an expert panel to provide professional opinions via anonymous questionnaires regarding organizational and technical factors influencing blockchain diffusion in healthcare. The study posed minimal risk, was not a patient-based study, and did not collect any personally identifiable participant information or sensitive health data. Therefore, ethical approval was determined to be unnecessary by the Office of the Director (Research & Extension), Khulna University of Engineering & Technology, Khulna-9203, Bangladesh. The expert recruitment period spanned from April 1, 2024 to July 30, 2024. All participants

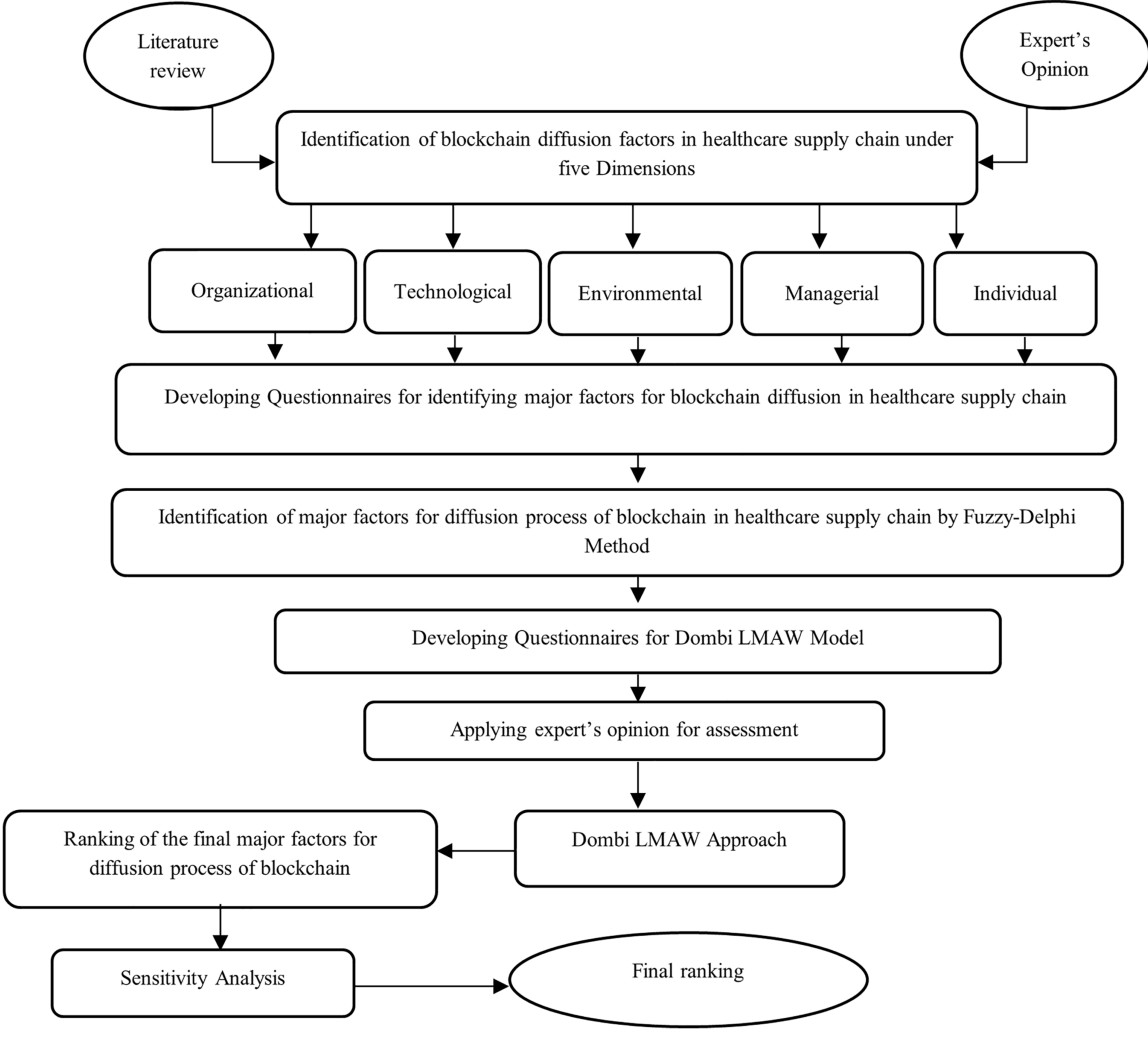

**Fig 1. The methodological framework.**

(aged 18 and up) provided informed written consent electronically, confirming their understanding of the study's purpose and confidentiality protocols. Anonymity of all responses was maintained throughout the data collection and analysis.

## Data collection procedure

The data collection was executed in two distinct stages aligned with the FDM and Dombi LMAW model. Participants were initially contacted and onboarded by the authors via telephone and email. The primary data collection instrument was a structured Google Form questionnaire, which was electronically distributed to all 10 experts. This form contained comprehensive instructions regarding the linguistic scales and the specific manner in which the criteria should be evaluated. All participants completed the questionnaire on the MCDM techniques. The comprehensive list of the initial 26 Blockchain Diffusion Factors identified for expert evaluation in the Fuzzy Delphi Method is shown in S1 Table in S1 File. The estimated time required for each participant to complete the questionnaires was approximately forty to forty-five minutes. The entire data gathering process took an approximate of four months to accomplish.

**Table 1. Overview of the experts' profiles.**

| Expert serial No. | Designation | Work Experience | Functional Role | Organizational Type | Selection Basis |
|---|---|---|---|---|---|
| E1 | Senior Doctor | 10 years | Clinical | Healthcare (Private Hospital) | Selected for senior clinical perspective on workflows and supply chain needs. |
| E2 | Junior Doctor | 3 years | Clinical | Healthcare (Govt. Hospital) | Selected for operational perspective on new technologies in healthcare. |
| E3 | Senior Nurse | 7 years | Clinical | Healthcare (Private Hospital) | Selected for logistical perspective from a nursing and coordination standpoint. |
| E4 | Junior Nurse | 4 years | Clinical | Healthcare (Private Hospital) | Selected for ground-level perspective on patient care and actual supply usage. |
| E5 | Hospital Manager | 5 years | Managerial | Healthcare (Govt. Hospital) | Selected for managerial perspective on integrating technology and logistics. |
| E6 | Supply Chain Manager | 6 years | Managerial | Healthcare (Govt. Hospital) | Selected for direct supply chain expertise in procurement and inventory. |
| E7 | IT Specialist | 5 years | Technical | Healthcare (Private Hospital) | Selected for technical implementation perspective as an IT specialist. |
| E8 | Pharmacist | 4 years | Technical (Clinical) | Healthcare (Pharmaceutical) | Selected for specialized technical expertise in medicine supply and inventory. |
| E9 | Academic Researcher | 8 years | Policy | Academic | Selected for subject matter expertise in blockchain technology and analytics. |
| E10 | Govt. Health Official | 6 years | Policy | Government | Selected for policy and regulatory perspective on public health supply chains. |

## Data validity and questionnaire reliability test

To construct the determination of the relevance of the blockchain diffusion factors in the healthcare supply chain, the fuzzy Delphi Method was used to obtain 10 expert responses. One of them represented the pharmaceutical field, two were connected with hospital supply chain and logistics management, three had experience in healthcare operations and technology (including clinical workflows, patient care, and coordination), two were affiliated with an academic or research institution with a focus on blockchain or analytics, one was an IT Healthcare operations manager, and one was a government health expert experienced with public health supply and policy. The survey included 30 questions; 26 questions were associated with the given factors and 4 questions were the demographic questions designed to describe the professional background and level of experiences of the respondent. Cronbach alpha was used to predict the internal consistency and reliability of the instrument and the measure was 0.812, which is better than the set standard of 0.7 and shows good level of reliability [33]. The data analysis was done by processing it using IBM SPSS statistics (Version 22.0). The non response bias was to be examined by doing chi-square analysis with the responses being categorized as early submissions and late submissions coded according to the time that they were completed. Statistically, the outcomes displayed greater similarities between the two groups, and thus a contribution of nonresponse bias was not evident, since the data specified by the collected expert opinions could be substantiated efficiently [43].

## Fuzzy Delphi Method

The Fuzzy Delphi Method (FDM) is the modified variant of Delphi that considers the linguistic preferences of people regarding decision making. This it done by combining fuzzy theory with the conventional Delphi. It takes the opinions of the experts and uses linguistic variables each of which is one of the following: very high, high, moderate, low and very low in a number of rounds of anonymous questionnaires and answers. This method then employs fuzzy set procedures to aggregate such responses after these language variables are converted into fuzzy sets using fuzzy membership functions [44]. The fuzzy Delphi approach is explained briefly in the following:

## Establishing the linguistic variables' triangular fuzzy number

A fuzzy set with a triangle membership function is called a triangular fuzzy set. The following is the equation for a triangular fuzzy set:

$$\mu_b(y) = \begin{cases} \frac{y-b_1}{b_2-b_1} & ; if\ b_1 \leq y \leq b_2 \\ \frac{b_3-y}{b_3-b_2} & ; if\ b_2 \leq y \leq b_3 \\ 0 & ; Otherwise \end{cases}$$

(1)

A triangular fuzzy number $\widetilde{b}_{ij}$ is defined as $\widetilde{b}_{ij} = a_{ij},\ b_{ij},\ c_{ij}$ in which $i \in \{1, 2, \dots n\}$ denotes the experts and $j \in \{1, 2, \dots m\}$ denotes the factors, where n and m are the number of experts and factors respectively. The fuzzy number set of the linguistic variables is obtained from the research framework presented in Table 2 [45].

## Collecting experts' opinions

After the identification of the factors, a questionnaire containing the linguistic factors provided in the above table is administered to four experts from academia and industry to measure the importance of the factors. Fuzzy triangular numbers in this study are applied in measuring the various factors. The method also utilizes a geometric mean method to determine the group choice by the experts [46].

## Defuzzification

The procedure for converting a fuzzy set into a suitable crisp set is called defuzzification. This method is often used when using fuzzy logic controllers. It involves selection of a defuzzification method depending on the problem-solving system of the actual problem being solved, mean-max, centroid, maximum, or weighted average. This procedure is required in order to convert the rather general control signals into definite values [47].

The geometric mean aggregation of "n" expert opinions results in a single aggregated TFN $\widetilde{B}_j = (a_j,\ b_j,\ c_j)$ for each factor j. The components are calculated using the geometric mean of the lower, median, and upper bounds of the individual TFNs provided by the experts:

$$\widetilde{B}_j = (a_j,\ b_j,\ c_j) = \left( \sqrt[n]{\prod_{i=1}^{n} a_{ij}}\ ,\ \sqrt[n]{\prod_{i=1}^{n} b_{ij}}\ ,\ \sqrt[n]{\prod_{i=1}^{n} c_{ij}} \right)$$

(2)

Table 2. Establishing the Fuzzy Number Set for Linguistic Variables.

| Linguistic Variables | Rating | Triangular Number $\widetilde{b}_{ij}$ |
|---|---|---|
| Very poor (VP) | 1 | (0,0,1) |
| Poor (P) | 2 | (1,2,3) |
| Medium Poor (MP) | 3 | (2,3,4) |
| Fair (F) | 4 | (3,4,5) |
| Medium good (MG) | 5 | (4,5,6) |
| High | 6 | (5,6,7) |
| High-very High | 7 | (6,7,8) |
| Very High | 8 | (7,8,9) |
| Very High-Extreme | 9 | (8,9,10) |
| Extreme | 10 | (10,10,10) |

Where $a_j = min_{i \in \{1,...,n\}} \{a_{ij}\}$ and $c_j = max_{i \in \{1,...,n\}} \{c_{ij}\}$

In this study, the expert group choice is ascertained through the application of a geometric mean model. The fuzzy weights are defuzzified using the simple center of gravity approach to yield a crisp value ($CV_j$), which may be calculated using the following formula:

$$CV_j = \frac{a_j + 4b_j + c_j}{6} \tag{3}$$

$i \in \{1, 2, ...., n\}$ denotes the experts and $j \in \{1, 2, ...., m\}$ denotes the factors.

## Determination of significant factors

The final step in the fuzzy Delphi technique is to identify the substantial factors, which is done by comparing the weight of each factor to the "S" threshold. The value of is calculated by averaging the weight of each factor. The screening principle is as follows:

$$S = \frac{\sum_{j=1}^{m} CV_j}{m} \tag{4}$$

Factor is chosen if $CV_j \geq S$.

Factor is rejected if $CV_j < S$.

To be compared, the values need to be transformed into crisp values. Following the defuzzification process, the consensus threshold, S was calculated as the average of the crisp values ($CV_j$) for all 26 initial factors. The determined threshold value used for screening was S = 6.9, as calculated in S3 Table in S1 File. The screening principle was then applied: a factor was retained for the second stage of the study if its crisp value met or exceeded the threshold ($CV_j \geq S$), indicating strong expert consensus on its significance. Conversely, any factor with a score below the threshold ($CV_j < S$) was eliminated as being contextually less relevant to the Bangladeshi healthcare sector. Applying this logic reduced the initial list from 26 factors to 22 significant factors. This filtering process ensures that the subsequent Dombi LMAW prioritization focuses exclusively on high-impact variables, thereby enhancing the analytical precision of the model. Fig 2 illustrates the Fuzzy Delphi Methodology for achieving consensus in healthcare supply chain concerns.

## Dombi Operator based LMAW

This section provided the preliminaries and basic operations of Dombi T-norms/T-conorms under triangular fuzzy numbers (TRNs).

## Dombi T-norm and T-conorm

In this study, triangular fuzzy numbers are applied to address the uncertainty in the information [48,49].

The notions and operations of the Dombi T-norm and T-conorm were introduced which present the advantage of good flexibility with the operational parameter [50]. Some definitions of the Dombi T-norm and T-conorm under triangular fuzzy numbers (TFNs) are expressed by:

**Definition 1.** Let $\alpha_1$ and $\alpha_2$ be any two real numbers. Then, the operations of Dombi T-norm and T-conorm between $\alpha_1$ and $\alpha_2$ are expressed by:

$$\check{T}_D(\alpha_1, \alpha_2) = \frac{1}{1 + \left\{ \left( \frac{1-\alpha_1}{\alpha_1} \right)^{\Upsilon} + \left( \left( \frac{1-\alpha_2}{\alpha_2} \right)^{\Upsilon} \right) \right\}^{\frac{1}{\Upsilon}}} \tag{5}$$

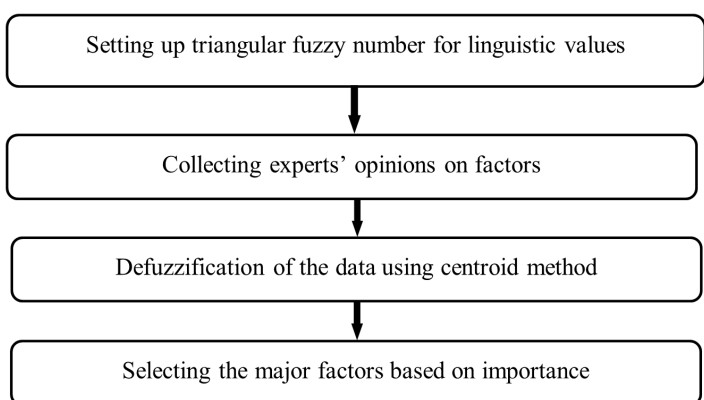

Fig 2. **Methodology of Fuzzy Delphi method.**

$$\check{T}_D{}^c\left(\alpha_1,\ \alpha_2\right) = 1 - \frac{1}{1 + \left\{\left(\left(\frac{1-\alpha_1}{\alpha_1}\right)^{\Upsilon} + \left(\left(\frac{1-\alpha_2}{\alpha_2}\right)^{\Upsilon}\right)\right)\right\}^{\frac{1}{\Upsilon}}}$$

(6)

Where $\Upsilon > 0$ and $(\alpha_1,\ \alpha_2) \in [0,1]$.

**Definition 2.** Let $\alpha_1 = \left(\alpha_1{}^{(a)}, \alpha_1{}^{(b)}, \alpha_1{}^{(c)}\right)$, and $\alpha_2 = \left(\alpha_2{}^{(a)}, \alpha_2{}^{(b)}, \alpha_2{}^{(c)}\right)$ be two TFNs, $\Upsilon > 0$ and let it be

$$f\left(\alpha_i\right) = \left(f\left(\alpha_i{}^{(a)}\right),\ f\left(\alpha_i{}^{(b)}\right),\ f\left(\alpha_i{}^{(c)}\right)\right) = \left(\frac{\alpha_i{}^{(a)}}{\sum_{i=1}^{n}\frac{\alpha_i{}^{(a)},\ \frac{\alpha_i{}^{(b)}}{\sum_{i=1}^{n}\alpha_i{}^{(b)}},\ \alpha_i{}^{(c)}}{\sum_{i=1}^{n}\alpha_i{}^{(c)}}}\right)$$ a fuzzy function, then some operational laws of TFNs based

on the Dombi T-norm and T-conorm are defined by:

1) Addition of two fuzzy numbers $\alpha_1$ and $\alpha_2$ can be defined as follows:

$$\alpha_1 + \alpha_2 = \left(\begin{array}{c} \sum_{i=1}^{2}\left(\alpha_i{}^{(a)}\right) - \left(\frac{\sum_{i=1}^{2}\left(\alpha_i{}^{(a)}\right)}{1 + \left\{\left(\frac{f\left(\alpha_1{}^{(a)}\right)}{1-f\left(\alpha_1{}^{(a)}\right)}\right)^{\Upsilon} + \left(\left(\frac{f\left(\alpha_2{}^{(a)}\right)}{1-f\left(\alpha_2{}^{(a)}\right)}\right)^{\Upsilon}\right)\right\}^{\frac{1}{\Upsilon}}}\right), \\[2em] \sum_{i=1}^{2}\left(\alpha_i{}^{(b)}\right) - \left(\frac{\sum_{i=1}^{2}\left(\alpha_i{}^{(b)}\right)}{1 + \left\{\left(\frac{f\left(\alpha_1{}^{(b)}\right)}{1-f\left(\alpha_1{}^{(b)}\right)}\right)^{\Upsilon} + \left(\left(\frac{f\left(\alpha_2{}^{(b)}\right)}{1-f\left(\alpha_2{}^{(b)}\right)}\right)^{\Upsilon}\right)\right\}^{\frac{1}{\Upsilon}}}\right), \\[2em] \sum_{i=1}^{2}\left(\alpha_i{}^{(c)}\right) - \left(\frac{\sum_{i=1}^{2}\left(\alpha_i{}^{(c)}\right)}{1 + \left\{\left(\frac{f\left(\alpha_1{}^{(c)}\right)}{1-f\left(\alpha_1{}^{(c)}\right)}\right)^{\Upsilon} + \left(\left(\frac{f\left(\alpha_2{}^{(c)}\right)}{1-f\left(\alpha_2{}^{(c)}\right)}\right)^{\Upsilon}\right)\right\}^{\frac{1}{\Upsilon}}}\right) \end{array}\right)$$

(7)

2) Multiplication of two fuzzy numbers $\alpha_1$ and $\alpha_2$ can be defined as follows:

$$\alpha_1 \times \alpha_2 = \left( \left( \frac{\sum_{i=1}^{2}\left(\alpha_i^{(a)}\right)}{1+\left\{\left(\left(\frac{1-f\left(\alpha_1^{(a)}\right)}{f\left(\alpha_1^{(a)}\right)}\right)^{\Upsilon}+\left(\frac{1-f\left(\alpha_2^{(a)}\right)}{f\left(\alpha_2^{(a)}\right)}\right)^{\Upsilon}\right\}^{\frac{1}{\Upsilon}}\right)}\right), \left( \frac{\sum_{i=1}^{2}\left(\alpha_i^{(b)}\right)}{1+\left\{\left(\frac{1-f\left(\alpha_1^{(b)}\right)}{f\left(\alpha_1^{(b)}\right)}\right)^{\Upsilon}+\left(\frac{1-f\left(\alpha_2^{(b)}\right)}{f\left(\alpha_2^{(b)}\right)}\right)^{\Upsilon}\right\}^{\frac{1}{\Upsilon}}}\right), \left( \frac{\sum_{i=1}^{2}\left(\alpha_i^{(c)}\right)}{1+\left\{\left(\frac{1-f\left(\alpha_1^{(c)}\right)}{f\left(\alpha_1^{(c)}\right)}\right)^{\Upsilon}+\left(\left(\frac{1-f\left(\alpha_2^{(c)}\right)}{f\left(\alpha_2^{(c)}\right)}\right)^{\Upsilon}\right)\right\}^{\frac{1}{\Upsilon}}}\right) \right) \quad (8)$$

3) Scalar Multiplication,

$$\kappa\alpha_1 = \left( \alpha_1^{(a)}.\left[1+\frac{\kappa}{\left(\frac{f\left(\alpha_1^{(a)}\right)}{1-f\left(\alpha_1^{(a)}\right)}\right)^{\Upsilon}}\right]^{\frac{-1}{\Upsilon}}, \alpha_1^{(b)}.\left[1+\frac{\kappa}{\left(\frac{f\left(\alpha_1^{(b)}\right)}{1-f\left(\alpha_1^{(b)}\right)}\right)^{\Upsilon}}\right]^{\frac{-1}{\Upsilon}}, \alpha_1^{(c)}.\left[1+\frac{\kappa}{\left(\frac{f\left(\alpha_1^{(c)}\right)}{1-f\left(\alpha_1^{(c)}\right)}\right)^{\Upsilon}}\right]^{\frac{-1}{\Upsilon}} \right) \quad (9)$$

4) Power,

$$\alpha_1^{\kappa} = \left( \alpha_1^{(a)}.\left[1+\frac{\kappa}{\left(\frac{1-f\left(\alpha_1^{(a)}\right)}{f\left(\alpha_1^{(a)}\right)}\right)^{\Upsilon}}\right]^{\frac{-1}{\Upsilon}}, \alpha_1^{(b)}.\left[1+\frac{\kappa}{\left(\frac{1-f\left(\alpha_1^{(b)}\right)}{f\left(\alpha_1^{(b)}\right)}\right)^{\Upsilon}}\right]^{\frac{-1}{\Upsilon}}, \alpha_1^{(c)}.\left[1+\frac{\kappa}{\left(\frac{1-f\left(\alpha_1^{(c)}\right)}{f\left(\alpha_1^{(c)}\right)}\right)^{\Upsilon}}\right]^{\frac{-1}{\Upsilon}} \right) \quad (10)$$

$\kappa>0$

**Definition 3.** Let $\alpha_j = \left(\alpha_j^{(a)}, \alpha_j^{(b)}, \alpha_j^{(c)}\right)$; $j = (1, 2, \ldots, n)$, a set of TFNs, and $\psi_j \in [0, 1]$ denotes the weight of coefficients of $\alpha_j$, which fulfills the requirement that it is $\sum_{j=1}^{n} \psi_j = 1$. Then, the fuzzy weighted averaging (FWA) operator and fuzzy weighted geometric averaging (FWGA) operator are expressed by:

$$FWA\left(\alpha_1, \alpha_2, \ldots, \alpha_n\right) = \sum_{j=1}^{n}\psi_j.\alpha_j = \left( \sum_{j=1}^{n}\psi_j.\alpha_j^{(a)}, \sum_{j=1}^{n}\psi_j.\alpha_j^{(b)}, \sum_{j=1}^{n}\psi_j.\alpha_j^{(c)} \right) \quad (11)$$

$$FWA\left(\alpha_1, \alpha_2, \ldots, \alpha_n\right) = \prod_{j=1}^{n}\left(\alpha_j\right)^{\psi_j} = \left( \prod_{j=1}^{n}\left(\alpha_j^{(a)}\right)^{\psi_j}, \prod_{j=1}^{n}\left(\alpha_j^{(b)}\right)^{\psi_j}, \prod_{j=1}^{n}\left(\alpha_j^{(c)}\right)^{\psi_j} \right) \quad (12)$$

## Determining Factors weights — fuzzy Dombi-based LMAW

The Dombi based LMAW methods are used to determine the weights of factors. The proposed Dombi based LMAW follows two consecutive stages.

**(1) Framework definition**

Step 1. The objectives of the decision-making criteria are explained, along with the factors that will be chosen and the experts that will be assembled to construct the suggested model. The Factors $?_j = (?_1, ?_2, \ldots, ?_n)$ $(j = 1, 2, \ldots, m)$ are evaluated by a set of experts $E_l = (E_1, E_2, \ldots, E_e)$ $(l = 1, 2, \ldots, e)$.

Step 2. The linguistic terms and their corresponding values are determined.

**(2) Determining Factors weights – DOMBI LMAW**

Step 3. The steps of this model are summarized as follows:

Step 3.1. Firstly, the fuzzy priority vector is described. The factors are evaluated by the experts with the help of fuzzy linguistic terms and their corresponding values (see Table 1). Then, the fuzzy priority vector $\chi^l = (\chi^l_1, \chi^l_2, \ldots, \chi^l_m)$ is obtained, where $(1 \leq \ell \leq e)$.

Step 3.2. The absolute anti-ideal point $(\theta_{AIP})$ is determined.

$$\theta_{AIP} < \min(\chi^l_j, \chi^l_j, \ldots, \chi^l_j) \tag{13}$$

The relationship between the fuzzy priority vector elements and the $\theta_{AIP}$ is found by:

$$\eta_j = \frac{\chi_j}{\theta_{AIP}} = \left( \frac{\chi_j^{(a)}}{\theta_{AIP}^{(c)}}, \frac{\chi_j^{(b)}}{\theta_{AIP}^{(b)}}, \frac{\chi_j^{(c)}}{\theta_{AIP}^{(a)}} \right) \tag{14}$$

Where $(\chi_j^{(a)}, \chi_j^{(b)}, \chi_j^{(c)})$ represents the elements of the priority vector $\eta$.

Step 3.3 The vectors of weight coefficients $\aleph_j$ are found by Eq. (15):

$$\aleph_j = \frac{\ln(\eta_j)}{\ln(\tau)} = \left( \frac{\ln(\eta_j^{(a)})}{\ln(\tau^{(c)})}, \frac{\ln(\eta_j^{(b)})}{\ln(\tau^{(b)})}, \frac{\ln(\eta_j^{(c)})}{\ln(\tau^{(a)})} \right) \tag{15}$$

Where $\tau = \prod_{j=1}^m \eta_j = \left( \prod_{j=1}^m \eta_j^{(a)}, \prod_{j=1}^m \eta_j^{(b)}, \prod_{j=1}^m \eta_j^{(c)} \right)$.

Step 3.4 The aggregated fuzzy priority vector is defined by:

$$\lambda_j = \lambda_j^{(a)}, \lambda_j^{(b)}, \lambda_j^{(c)}) = \left( \frac{\sum_{j=1}^m \left( \aleph_{ij}^{(a)} \right)}{1 + \left\{ \sum_{j=1}^m w_j \left( \frac{1 - f\left( \aleph_{ij}^{(a)} \right)}{f\left( \aleph_{ij}^{(a)} \right)} \right) \right\}^{\frac{1}{}}}, \frac{\sum_{j=1}^m \left( \aleph_{ij}^{(b)} \right)}{1 + \left\{ \sum_{j=1}^m w_j \left( \frac{1 - f\left( \aleph_{ij}^{(b)} \right)}{f\left( \aleph_{ij}^{(b)} \right)} \right) \right\}^{\frac{1}{}}}, \frac{\sum_{j=1}^m \left( \aleph_{ij}^{(c)} \right)}{1 + \left\{ \sum_{j=1}^m w_j \left( \frac{1 - f\left( \aleph_{ij}^{(c)} \right)}{f\left( \aleph_{ij}^{(c)} \right)} \right) \right\}^{\frac{1}{}}} \right) \tag{16}$$

Where $\sigma > 0$, and $\sum_{\updownarrow=1}^e w_l$.

Step 3.5 The triangular fuzzy value of $\lambda_j = \lambda_j^{(a)}, \lambda_j^{(b)}, \lambda_j^{(c)}$ is defuzzified by:

$$\text{def}(\lambda_j) = \frac{\lambda_j^{(a)} + 4\lambda_j^{(b)} + \lambda_j^{(c)}}{6} \tag{17}$$

The used model is proposed by [51]. The flow chart of the used model is shown in Fig 3.

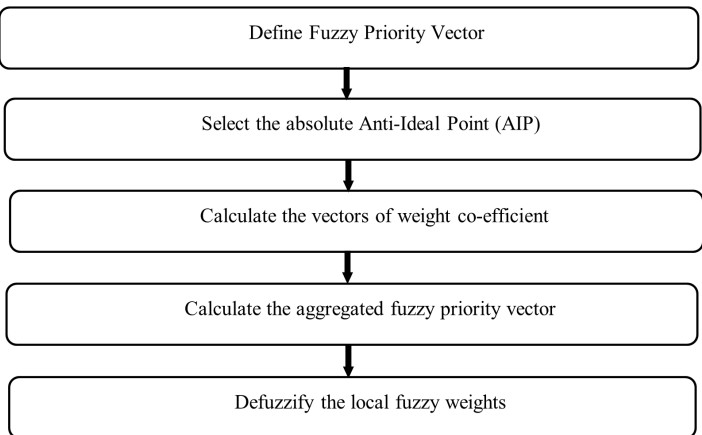

**Fig 3. Dombi LMAW method.**

## Case study

With an emphasis on strengthening resilience in the face of adversities like pandemics, this study sought to investigate the elements supporting the adoption of Blockchain technology within the supply chain management of the healthcare sector in Bangladesh. To evaluate the several aspects impacting the diffusion of Blockchain technology in healthcare supply chain management, firstly Fuzzy Delphi Method was utilized to identify the substantial factors, where the assessments of the factors are reported in S2 Table in S1 File and the defuzzified scores of the factors are provided in S3 Table in S1 File. Then a fuzzy Dombi-based decision-making model was utilized. In order to increase resilience, this case study focuses on the healthcare sector in Bangladesh and attempts to pinpoint the elements that make a blockchain implementation successful. Total four healthcare were examined for this scenario. The initial set of 26 factors was reduced to 22 significant factors through the Fuzzy Delphi screening process. The consensus threshold, S was calculated as the average of the defuzzified crisp values ($CV_j$) of all 26 initial factors, yielding S = 6.9. The four factors eliminated from the analysis, having been assessed by the expert panel as contextually less relevant (i.e., their $CV_j < S$), were: Lack of Practical Use Cases (OF6), Cybersecurity Risks (TF7), Performance Expectancy (EF5) and Effort Expectancy (IF4). The exclusion of these factors suggests that in the current resource-constrained healthcare landscape of Bangladesh, experts prioritize foundational organizational readiness and leadership support over technical secondary risks or individual behavioral expectations. This systematic reduction ensures that the subsequent Dombi LMAW prioritization focuses solely on the factors deemed most critical for the Bangladeshi healthcare context. The finalized list of blockchain diffusion factors obtained from Fuzzy Delphi method is shown in Table 3.

## Healthcare A

Healthcare A is a government-owned emergency ward hospital in Khulna, picked because it represents the scenario of the government-based healthcare activities in a big-city setup. The data gathered in this hospital was to get to know the issues associated with the state-run supply chain digitalization. The hospital had challenges because although it was staffed by experienced professionals, it was having challenges of old data systems, bad interoperability and poor organizational readiness. Using the Fuzzy Delphi and Dombi LMAW methodology, possible weaknesses were outlined (especially in stakeholder trust and infrastructure preparedness) and could be useful in the adoption of blockchain in a similar public institution by offering opportunities where action can be taken.

**Table 3. Finalized list of blockchain diffusion factors in healthcare sector.**

| Sl. | Factors | How does it affect blockchain diffusion? | Sources |
|---|---|---|---|
| 1 | Organizational readiness (OF1) | The technical ability, infrastructure and digital maturity within any organization is a key factor in an efficient realization of blockchain systems. Preparedness saves time of implementation and resistance. | [35] |
| 2 | Implementation cost (OF2) | Expensive infrastructure, training, and upgrade of the systems can also serve as a barrier to the readiness of organizations to test blockchain technologies. | [35] |
| 3 | Resistance to change (OF3) | The fear of employees, the lack of knowledge, and alienation of routines can turn into a phenomenon of resistance to new technologies, such as blockchain, in which case the change is slowed down. | [34] |
| 4 | Top management support (OF4) | Proper leadership commitment guarantees a strategic ledger, available resources and organization motivation on blockchain activities. | [35] |
| 5 | Lack of ROI clarity (OF5) | The uncertainties about the potential returns of the blockchain investment discourage organization to invest in the blockchain, particularly, when this investment is made with limited funds. | [33] |
| 6 | Blockchain scalability (TF1) | The number of transactions and users should be increasingly supported by blockchain systems. | [35] |
| 7 | Interoperability with legacy systems (TF2) | Interoperability with legacy systems means the seamless integration of blockchain products and that the adjustment will have minimal impact on operations and will not be costly in terms of extra IT investments. | [36] |
| 8 | Smart contract automation (TF3) | Smart contracts carry out and automate the process and business rules and enhance efficiency and trust. Nonetheless, technical complications and mistakes may be an obstacle. | [34] |
| 9 | Integration with IT infrastructure (TF4) | An efficient IT infrastructure means that blockchain platforms can easily interact with existing databases and enterprise systems, which raises their effectiveness. | [37] |
| 10 | Standardization issues (TF5) | The lack of standard processes and interfaces written on different blockchain systems results in incompatibility and will not allow wide-scale blockchain adoption. | [38] |
| 11 | Traceability (TF6) | Blockchain also improves traceability where products and transactions can be tracked in real time which is imperative in compliance and trust among stakeholders. | [10] |
| 12 | Competitive pressure (EF1) | Competitive pressure forces firms to seek new tools, such as blockchain, to gain efficiency, lower the prices, and stand out in the market. | [35] |
| 13 | Innovation pressure (EF2) | The pressure to be innovative and the needs in the market require organizations to use upcoming technology such as blockchain to remain competent in the market. | [34] |
| 14 | Vendor readiness (EF3) | The willingness and capabilities of a vendor in using blockchain techniques like technical skills will influence the accessibility of the supply chain ecosystem in general. | [39] |
| 15 | Regulatory and legal clarity (EF4) | Transparent rules bring a legal guarantee and the application of compliance regulations, which eliminates the risks and doubts that form the major motivation not to adopt blockchains. | [35] |
| 16 | Transparency and immutability (MF1) | The use of blockchain in business makes its records as such non-removable and transparent, making the operation more traceable and accountable, raising trust and compliance. | [40] |
| 17 | Perceived benefits (MF2) | Blockchain offers potential changes that are seen beneficial to firms such as transparency, decrease in fraud, and efficiency which makes them worthy investment and experimentation opportunities. | [10] |
| 18 | Trust among stakeholders (MF3) | Open data sharing and collaboration are essential components of successful decentralized blockchain ecosystems since such networks require the members to trust one another. | [40] |
| 19 | Disintermediation (MF4) | Removing the intermediaries through blockchain can hence reduce cost of making transactions, enhance speed of transactions as well as minimize dependency hence simplifying workflows. | [40] |
| 20 | Lack of awareness (IF1) | Lack of knowledge of blockchain capabilities by employees and decision-makers limits innovative activity, investment and implementation of pilot work. | [41] |
| 21 | Trust in technology (IF2) | The more convinced that blockchain is secure and reliable, the more users are willing to embrace its incorporation and application in business processes. | [33] |
| 22 | Social influence (IF3) | The social pressure of the peer, the competitors, and the society as a whole can help people to accept and promote blockchain technologies in their organization. | [33] |

A questionnaire is conducted to evaluate the Factors. Each criterion is scored by the experts. The questionnaires are sent to ten experts. The results are then gathered. Twenty-two Factors $A_j$ = (1, 2,..., 22) are evaluated by a set of experts $S_1$ = (1, 2,....,10) with the help of Table 4 which presents the linguistic scale to obtain the experts' opinions [52].

**Table 4. Fuzzy linguistic scale.**

| Linguistic terms | Membership function |
|---|---|
| Absolutely low (AL) | (1,1,1) |
| Very low (VL) | (1,2,3) |
| Low(L) | (2,3,4) |
| Medium low (ML) | (3,4,5) |
| Equal (E) | (4,5,6) |
| Medium high (MH) | (5,6,7) |
| High(H) | (6,7,8) |
| Very high (VH) | (7,8,9) |
| Absolutely high (AH) | (8,9,9) |

For Healthcare A, the experimental results are shown below:

**Step 1.** The fuzzy priority vectors are defined by ten experts in this step. The obtained values are given in Table 5.

**Step 2.** Using Eq. (13) and expressing the relationship vector, the value of AIP= (0.4, 0.5, 0.6) is integrated.

Later, AIP is applied to obtain the ratio vector by Eq. (14). The results are given in Table 6. For example, the ratio vector for OF1 is calculated by Eq. (14) as follows:

$$\left( \eta_1{}^{1(a)}, \eta_1{}^{1(b)}, \eta_1{}^{1(c)} \right) = \left( \frac{\chi_1{}^{1(a)}}{\theta_{AIP}^{(c)}}, \frac{\chi_1{}^{1(b)}}{\theta_{AIP}^{(b)}}, \frac{\chi_j{}^{1(c)}}{\theta_{AIP}^{(a)}} \right) = \left( \frac{7}{0.6}, \frac{8}{0.5}, \frac{9}{0.4} \right) = (11.67, \ 16, \ 22.5)$$

The data of the Factors Ratio Vectors for Healthcare A from the remaining five experts (E6-E10) is shown in S4 Table in S1 File.

**Step 3.** The final vectors of weighting coefficients are calculated in this step.

For example, the weighting coefficients for OF1 is calculated by Eq. (15) as follows:

$$\tau_1^1 = \prod_{j=1}^{22} \eta_1{}^1 = \left( \prod_{j=1}^{22} \eta_1{}^{1(a)}, \prod_{j=1}^{22} \eta_1{}^{1(b)}, \prod_{j=1}^{22} \eta_1{}^{1(c)} \right) = \left( 8.48 \times 10^{21}, 1.52 \times 10^{25}, 3.32 \times 10^{28} \right)$$

$$\aleph_1{}^1 = \frac{\ln\left( \eta_1{}^1 \right)}{\ln\left( \tau_1^1 \right)} = \left( \frac{\ln\left( \eta_1{}^{1(a)} \right)}{\ln\left( \tau_1{}^{1(c)} \right)}, \frac{\ln\left( \eta_1{}^{1(b)} \right)}{\ln\left( \tau_1{}^{1(b)} \right)}, \frac{\ln\left( \eta_1{}^{1(c)} \right)}{\ln\left( \tau_1{}^{1(a)} \right)} \right) = (0.03741, \ 0.047817, \ 0.061664).$$

**Step 4.** The aggregated fuzzy priority vector is calculated by Eq. (16). For example, the aggregated fuzzy priority vector for OF1 is,

$$\lambda_1^1 = \lambda_1^{1(a)}, \lambda_1^{1(b)}, \lambda_1^{1(c)}) =$$

$$\left( \frac{\sum_{j=1}^{22} \left( \aleph_1^{1(a)} \right)}{1 + \left\{ \sum_{j=1}^{22} w_1 \left( \frac{1 - f\left( \aleph_1^{1(a)} \right)}{f\left( \aleph_1^{1(a)} \right)} \right)^{\lambda} \right\}^{\frac{1}{\lambda}}}, \frac{\sum_{j=1}^{22} \left( \aleph_1^{1(b)} \right)}{1 + \left\{ \sum_{j=1}^{22} w_1 \left( \frac{1 - f\left( \aleph_1^{1(b)} \right)}{f\left( \aleph_1^{1(b)} \right)} \right)^{\lambda} \right\}^{\frac{1}{\lambda}}}, \frac{\sum_{j=1}^{22} \left( \aleph_1^{1(c)} \right)}{1 + \left\{ \sum_{j=1}^{22} w_1 \left( \frac{1 - f\left( \aleph_1^{1(c)} \right)}{f\left( \aleph_1^{1(c)} \right)} \right)^{\lambda} \right\}^{\frac{1}{\lambda}}} \right)$$

$$= \left( \frac{0.380211}{1 + \left\{ 0.045455(253.4692)^1 \right\}^{\frac{1}{1}}}, \frac{0.476439}{1 + \left\{ 0.045455(200.4942)^1 \right\}^{\frac{1}{1}}}, \frac{0.59645}{1 + \left\{ 0.045455(159.0191)^1 \right\}^{\frac{1}{1}}} \right)$$

$$= (0.030365, \ 0.047109, \ 0.072488)$$

**Table 5. Fuzzy Priority Vectors for Healthcare A.**

| Factors | E1 | E2 | E3 | E4 | E5 | E6 | E7 | E8 | E9 | E10 |
|---------|------|------|------|------|------|------|------|------|------|------|
| OF1 | (7,8,9) | (5,6,7) | (7,8,9) | (8,9,9) | (7,8,9) | (8,9,9) | (8,9,9) | (6,7,8) | (8,9,9) | (8,9,9) |
| OF2 | (7,8,9) | (6,7,8) | (6,7,8) | (8,9,9) | (4,5,6) | (6,7,8) | (8,9,9) | (7,8,9) | (7,8,9) | (8,9,9) |
| OF3 | (6,7,8) | (8,9,9) | (4,5,6) | (7,8,9) | (5,6,7) | (7,8,9) | (7,8,9) | (8,9,9) | (6,7,8) | (7,8,9) |
| OF4 | (8,9,9) | (7,8,9) | (7,8,9) | (7,8,9) | (7,8,9) | (8,9,9) | (8,9,9) | (7,8,9) | (8,9,9) | (8,9,9) |
| OF5 | (5,6,7) | (5,6,7) | (5,6,7) | (7,8,9) | (1,2,3) | (6,7,8) | (6,7,8) | (6,7,8) | (6,7,8) | (7,8,9) |
| TF1 | (7,8,9) | (6,7,8) | (6,7,8) | (6,7,8) | (7,8,9) | (8,9,9) | (8,9,9) | (7,8,9) | (7,8,9) | (6,7,8) |
| TF2 | (7,8,9) | (6,7,8) | (6,7,8) | (7,8,9) | (6,7,8) | (7,8,9) | (7,8,9) | (7,8,9) | (8,9,9) | (7,8,9) |
| TF3 | (7,8,9) | (4,5,6) | (5,6,7) | (5,6,7) | (1,2,3) | (6,7,8) | (8,9,9) | (5,6,7) | (6,7,8) | (5,6,7) |
| TF4 | (8,9,9) | (4,5,6) | (7,8,9) | (7,8,9) | (6,7,8) | (7,8,9) | (8,9,9) | (5,6,7) | (8,9,9) | (7,8,9) |
| TF5 | (4,5,6) | (6,7,8) | (7,8,9) | (8,9,9) | (4,5,6) | (2,3,4) | (5,6,7) | (7,8,9) | (8,9,9) | (8,9,9) |
| TF6 | (6,7,8) | (7,8,9) | (7,8,9) | (7,8,9) | (4,5,6) | (4,5,6) | (7,8,9) | (8,9,9) | (8,9,9) | (7,8,9) |
| EF1 | (5,6,7) | (6,7,8) | (5,6,7) | (7,8,9) | (8,9,9) | (7,8,9) | (6,7,8) | (7,8,9) | (6,7,8) | (7,8,9) |
| EF2 | (5,6,7) | (6,7,8) | (6,7,8) | (6,7,8) | (1,2,3) | (2,3,4) | (6,7,8) | (7,8,9) | (7,8,9) | (6,7,8) |
| EF3 | (4,5,6) | (5,6,7) | (7,8,9) | (8,9,9) | (6,7,8) | (5,6,7) | (5,6,7) | (6,7,8) | (8,9,9) | (8,9,9) |
| EF4 | (7,8,9) | (5,6,7) | (6,7,8) | (7,8,9) | (8,9,9) | (7,8,9) | (8,9,9) | (6,7,8) | (7,8,9) | (7,8,9) |
| MF1 | (5,6,7) | (6,7,8) | (7,8,9) | (7,8,9) | (6,7,8) | (7,8,9) | (5,6,7) | (7,8,9) | (8,9,9) | (8,9,9) |
| MF2 | (7,8,9) | (7,8,9) | (6,7,8) | (4,5,6) | (1,2,3) | (6,7,8) | (8,9,9) | (8,9,9) | (7,8,9) | (4,5,6) |
| MF3 | (7,8,9) | (7,8,9) | (6,7,8) | (7,8,9) | (6,7,8) | (7,8,9) | (8,9,9) | (8,9,9) | (8,9,9) | (6,7,8) |
| MF4 | (4,5,6) | (5,6,7) | (8,9,9) | (7,8,9) | (8,9,9) | (7,8,9) | (5,6,7) | (6,7,8) | (8,9,9) | (6,7,8) |
| IF1 | (5,6,7) | (6,7,8) | (5,6,7) | (4,5,6) | (4,5,6) | (5,6,7) | (6,7,8) | (7,8,9) | (6,7,8) | (4,5,6) |
| IF2 | (7,8,9) | (7,8,9) | (5,6,7) | (7,8,9) | (7,8,9) | (7,8,9) | (8,9,9) | (8,9,9) | (5,6,7) | (5,6,7) |
| IF3 | (6,7,8) | (6,7,8) | (7,8,9) | (7,8,9) | (8,9,9) | (5,6,7) | (7,8,9) | (7,8,9) | (7,8,9) | (7,8,9) |

Where, $w_1 = \frac{1}{22} = 0.045455$ and $\sigma = 1$

**Step 5.** The Scores are obtained by defuzzifying the aggregated local fuzzy weights by Eq. (17).

For example, the score of OF1 is,

$$\mathrm{def}\left(\lambda_1\right) = \frac{\lambda_1^{(a)} + 4\lambda_1^{(b)} + \lambda_1^{(c)}}{6} = \frac{0.030365 + 4 \times 0.047109 + 0.072488}{6} = 0.048548$$

Then, the ranks of the factors are determined from 1 to 22 through the descending order of the scores. The complete results of the prioritization for Healthcare A are shown in Table 7.

Fig 4 shows the final scores of Blockchain Success factors for diffusing in Healthcare A supply chain.

## Healthcare B

Healthcare B is located in Rajshahi and has been selected as a second example of the representative treating institution by the geographic area to gain comparative patterns about the regional differences in digital readiness. Regardless of the fame and magnitude, the hospital experienced a disjointed communication flow, unstandardized processes of supplies, and financial restrictions. To the extent that the inclusion of this site served to confirm that such heavyweight factors of diffusion as management support and regulatory clarity were indeed found to have a consistent importance across regions. The outcome of the study is bound to shape directional measures that will engage these common public sector challenges. Table 8 shows the final experimental results for Healthcare B.

**Table 6. Factors ratio vectors for Healthcare A.**

| Factors | E1 | E2 | E3 | E4 | E5 |
|---|---|---|---|---|---|
| OF1 | (11.67,16,22.5) | (8.33,12,11.67) | (11.67,16,22.5) | (13.33,18,22.5) | (11.67,16,22.5) |
| OF2 | (11.67,16,22.5) | (10,14,13.33) | (10,14,20) | (13.33,18,22.5) | (6.67,10,15) |
| OF3 | (10,14,20) | (13.33,18,15) | (6.67,10,15) | (11.67,16,22.5) | (8.33,12,17.5) |
| OF4 | (13.33,18,22.5) | (11.67,16,15) | (11.67,16,22.5) | (11.67,16,22.5) | (11.67,16,22.5) |
| OF5 | (8.33,12,17.5) | (8.33,12,11.67) | (8.33,12,17.5) | (11.67,16,22.5) | (1.67,4,7.5) |
| TF1 | (11.67,16,22.5) | (10,14,13.33) | (10,14,20) | (10,14,20) | (11.67,16,22.5) |
| TF2 | (11.67,16,22.5) | (10,14,13.33) | (10,14,20) | (11.67,16,22.5) | (10,14,20) |
| TF3 | (11.67,16,22.5) | (6.67,10,10) | (8.33,12,17.5) | (8.33,12,17.5) | (1.67,4,7.5) |
| TF4 | (13.33,18,22.5) | (6.67,10,10) | (11.67,16,22.5) | (11.67,16,22.5) | (10,14,20) |
| TF5 | (6.67,10,15) | (10,14,13.33) | (11.67,16,22.5) | (13.33,18,22.5) | (6.67,10,15) |
| TF6 | (10,14,20) | (11.67,16,15) | (11.67,16,22.5) | (11.67,16,22.5) | (6.67,10,15) |
| EF1 | (8.33,12,17.5) | (10,14,13.33) | (8.33,12,17.5) | (11.67,16,22.5) | (13.33,18,22.5) |
| EF2 | (8.33,12,17.5) | (10,14,13.33) | (10,14,20) | (10,14,20) | (1.67,4,7.5) |
| EF3 | (6.67,10,15) | (8.33,12,11.67) | (11.67,16,22.5) | (13.33,18,22.5) | (10,14,20) |
| EF4 | (11.67,16,22.5) | (8.33,12,11.67) | (10,14,20) | (11.67,16,22.5) | (13.33,18,22.5) |
| MF1 | (8.33,12,17.5) | (10,14,13.33) | (11.67,16,22.5) | (11.67,16,22.5) | (10,14,20) |
| MF2 | (11.67,16,22.5) | (11.67,16,15) | (10,14,20) | (6.67,10,15) | (1.67,4,7.5) |
| MF3 | (11.67,16,22.5) | (11.67,16,15) | (10,14,20) | (11.67,16,22.5) | (10,14,20) |
| MF4 | (6.67,10,15) | (8.33,12,11.67) | (13.33,18,22.5) | (11.67,16,22.5) | (13.33,18,22.5) |
| IF1 | (8.33,12,17.5) | (10,14,13.33) | (8.33,12,17.5) | (6.67,10,15) | (6.67,10,15) |
| IF2 | (11.67,16,22.5) | (11.67,16,15) | (8.33,12,17.5) | (11.67,16,22.5) | (11.67,16,22.5) |
| IF3 | (10,14,20) | (10,14,13.33) | (11.67,16,22.5) | (11.67,16,22.5) | (13.33,18,22.5) |

In the prioritization of the factors across the healthcare organizations, Healthcare B has experienced mild changes in ranks relative to Healthcare A: there is interchange of ranks between TF5 (Standardization Issues) and IF1 (Lack of awareness) where TF5 ranks 18th and IF1 ranks 17th in Healthcare B. There is also interchange between MF1 (Transparency and immutability) and IF2 (Trust in technology) in which MF1 is 11th and IF2 is 10th. Such shifts signify regional and contextual differences in the way technological, managerial and individual aspects are understood and valued within the healthcare supply chain.

## Healthcare C

Healthcare C is a leading middle-sized privately owned hospital in Khulna. It was chosen to give an understanding of how moderately scaled independent institutions treat the assimilation of the emerging technology. Although this facility has relative flexibility in its operations, it has also been facing challenges like mistrust and resistance to change by the stakeholders. Data gathered in this hospital enhanced the analysis by showing how even in well-equipped surroundings non-technical barriers are still present. The results of the prioritization advised the hospital to initiate regularized awareness and training to smoothen the road to blockchain adoption. Table 9 shows the final experimental results for Healthcare C.

In Healthcare C, the placement of TF1 (Blockchain Scalability) and TF2 (Interoperability with Legacy Systems) have swapped meaning that the priority has shifted in the perceived importance of the technical infrastructure concerns. Likewise, EF4 (Regulatory and Legal Clarity) and IF3 (Social Influence), have also exchanged places which implies that stakeholders' social dynamics are regarded to be a bit more relevant than regulatory clarity as far as blockchain uptake is concerned.

**Table 7. The local fuzzy values, scores and ranks of the factors for Healthcare A.**

| Factors | Fuzzy value | Score def $(\lambda_j)$ | Rank |
|---|---|---|---|
| OF1 | (0.03,0.05,0.07) | 0.0485 | 2 |
| OF2 | (0.03,0.04,0.07) | 0.0464 | 8 |
| OF3 | (0.03,0.04,0.07) | 0.0456 | 12 |
| OF4 | (0.03,0.05,0.07) | 0.0501 | 1 |
| OF5 | (0.02,0.04,0.06) | 0.0380 | 20 |
| TF1 | (0.03,0.05,0.07) | 0.0473 | 5 |
| TF2 | (0.03,0.05,0.07) | 0.0475 | 4 |
| TF3 | (0.02,0.04,0.06) | 0.0368 | 21 |
| TF4 | (0.03,0.04,0.07) | 0.0463 | 9 |
| TF5 | (0.02,0.04,0.06) | 0.0411 | 17 |
| TF6 | (0.03,0.04,0.07) | 0.0454 | 14 |
| EF1 | (0.03,0.04,0.07) | 0.0455 | 13 |
| EF2 | (0.02,0.03,0.06) | 0.0362 | 22 |
| EF3 | (0.03,0.04,0.07) | 0.0440 | 16 |
| EF4 | (0.03,0.05,0.07) | 0.0472 | 6 |
| MF1 | (0.03,0.04,0.07) | 0.0462 | 10 |
| MF2 | (0.02,0.04,0.06) | 0.0389 | 19 |
| MF3 | (0.03,0.05,0.07) | 0.0481 | 3 |
| MF4 | (0.03,0.04,0.07) | 0.0449 | 15 |
| IF1 | (0.02,0.04,0.06) | 0.0399 | 18 |
| IF2 | (0.03,0.04,0.07) | 0.0462 | 11 |
| IF3 | (0.03,0.05,0.07) | 0.0469 | 7 |

## Healthcare D

Healthcare D is one of the high-end, exclusive medical service providers, located in Khulna as well. It was added to understand how even technologically advanced organizations struggle with the ability to ensure that blockchain investments support the perceived revenue and internal capacity. Including this hospital in the decision enabled the research to grasp the fact that digital maturity is not always characterized by blockchain adoption readiness. The knowledge acquired served to reinforce the value of leadership commitment and understanding the return on investment, especially where the business viability is at stake, like in the private sector. The final experimental result for Healthcare D is shown in Table 10.

In Healthcare D, only a significant change in factor ranking is the swap between MF1 (Transparency and Immutability) and OF2 (Implementation Cost), where MF1 moves up to 8th position and OF2–10th. Such transition indicates that in such an auxiliary high level of privatization in the private healthcare industry, the value of transparency and immutability of blockchain properties overshadows higher-order cost efficiency, another indication of putting long-term working integrity before a more short-term cost-driven paradigm. The rest of the factors retain their former rankings.

## Sensitivity Analysis

In this section, the analysis of the parameter σ based on the subjective evaluations of the experts in the decision-making problem was made. 50 experiments were carried out for σ in fuzzy Dombi parameter to calculate the factors weights. It was observed if the ranking of the factors were changed in changing the value of σ that is from the interval $1 \leq \sigma \leq 50$. Table 11 represents the final score and ranking of the Healthcare A with new value of σ, which is 2.

Fig 5 illustrates the change in the value of σ, and the new scores and ranking of the factors for the Healthcare A.

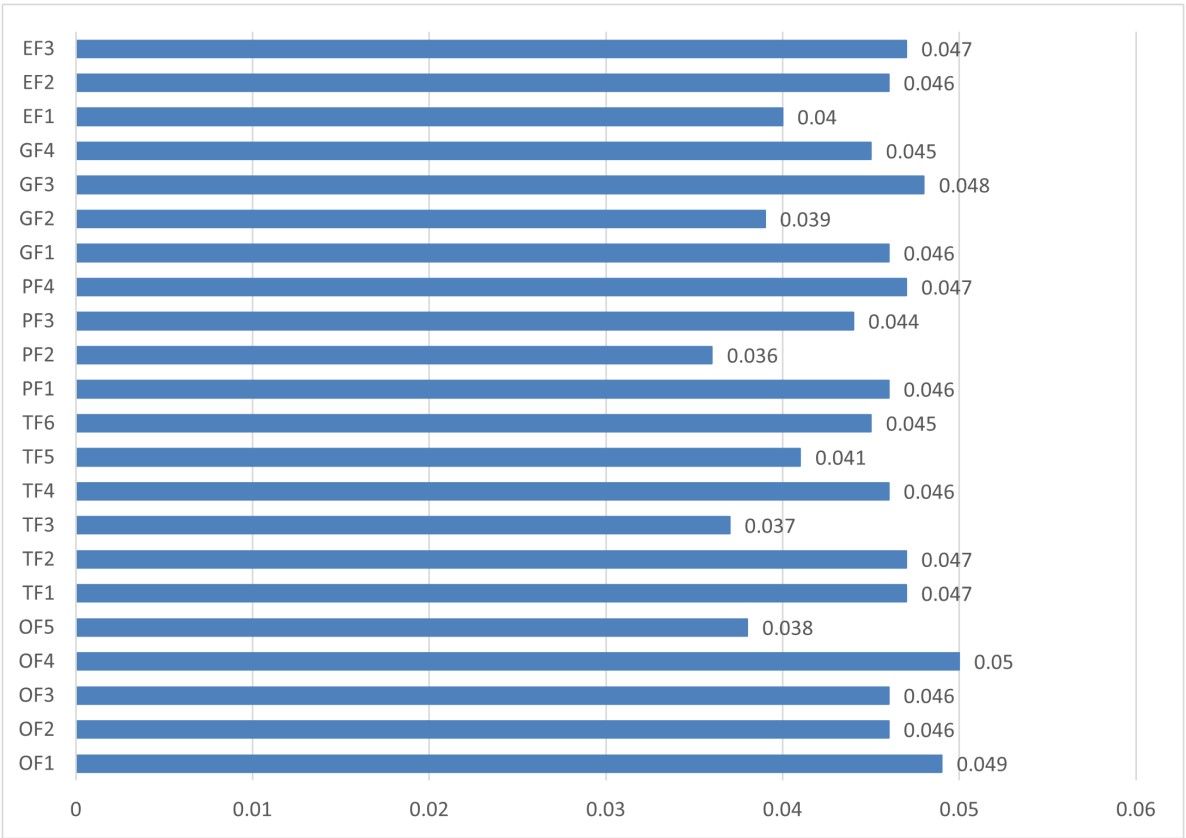

**Fig 4. Final scores of Blockchain success factors in Healthcare A.**

## Discussion

The study adopted a two-stage process which was structured as Fuzzy Delphi Method and Dombi Logarithmic Mean Aggregated Weighting (Dombi LMAW) to identify and rank the critical factors that affected blockchain diffusion in the Bangladeshi healthcare supply chain. This process started by identifying 26 possible sub-factors by conducting an extensive literature and empirical studies review across the field of technology, organization, management, environment and in the case of individual level considerations. These factors were then put through expert rating with the help of the Fuzzy Delphi Method so that contextual validity and expert agreement could manifest. Ten experts with different yet relevant background such as pharmaceutical manufacturing, hospital logistics, healthcare IT service providers, academia, government of public health service, and digital supply chain management took part in the assessment. A structured questionnaire was given to the experts according to which one factor was rated against another by uniform terms of linguistic nature which were turned into later triangular fuzzy number. The defuzzified aggregated fuzzy scores of each of these factors were then used to get the threshold values. A threshold cut-off value was generated using the standard Delphi procedure to provide the minimum level of consensus that should be met in order to remain a factor [53]. When the mean score of a factor became less than this level, it was considered to be less important and hence to be omitted in further analysis. Four factors were discarded in the process, one pertaining to the technological category, one in organizational domain, one in managerial arena, and one in the individualistic dimension. These factors that were eliminated were evaluated to earn contextual pertinence or less powerful effect in the healthcare blockchain ecosystem in Bangladesh as shown by their low

**Table 8. The fuzzy scores and ranks of the factors for Healthcare B.**

| Factor | Score | Rank |
|---|---|---|
| OF1 | 0.0496 | 2 |
| OF2 | 0.0466 | 8 |
| OF3 | 0.0457 | 12 |
| OF4 | 0.0512 | 1 |
| OF5 | 0.0386 | 20 |
| TF1 | 0.0472 | 5 |
| TF2 | 0.0479 | 4 |
| TF3 | 0.0368 | 21 |
| TF4 | 0.0465 | 9 |
| TF5 | 0.0411 | 18 |
| TF6 | 0.0455 | 14 |
| EF1 | 0.0456 | 13 |
| EF2 | 0.0361 | 22 |
| EF3 | 0.0448 | 16 |
| EF4 | 0.0470 | 6 |
| MF1 | 0.0462 | 11 |
| MF2 | 0.0395 | 19 |
| MF3 | 0.0488 | 3 |
| MF4 | 0.0451 | 15 |
| IF1 | 0.0425 | 17 |
| IF2 | 0.0463 | 10 |
| IF3 | 0.0469 | 7 |

defuzzied mean scores and non-consensus between specialists. This data filtration meant that the 22 most significant sub-factors were to be retained and used in the next prioritizing step, which enhanced the transparent nature and analytical accuracy of the model.

In order to rank the 22 short-listed factors, the Dombi logarithmic Mean Aggregated Weighting (Dombi LMAW) method was utilized to incorporate the flexibility of Dombi operator and the strength of normalization property of Logarithmic mean aggregation. Experts were requested to give the pair-wise comparisons of every factor according to their perceived relative importance of each factor to influence the blockchain diffusion in the healthcare supply chain. The Dombi operator, which uses a parametric control to alter compensation between elements, was used for these comparisons. It helps create a more nuanced prioritization where it aims to identify the interdependencies between the factors and posits expert uncertainty [50]. These judgments were then combined into normalized weights using logarithmic mean to give a final rank of each factor. The outcomes indicated that Top Management Support (fuzzy score = 0.0501), Organizational Readiness (0.0485), Trust Among Stakeholders (0.0481) and Interoperability with legacy systems (0.0475) were found to be the most influential factors. These highest scoring factors are indicative of how paramount leadership, stakeholder cooperation, and integration of the system are in implementing blockchain. On the other hand, factors such as Smart Contract Automation (0.0368) and Innovation Pressure (0.0362) were on the lowest position on the list and this means that more developed or externally imposed innovations are not that urgent when compared to the basic organizational needs and infrastructural issues in case of the Bangladesh healthcare sector. The factor priority determined by Dombi LMAW establishes the optimal sequence for resource allocation and the diffusion effect is achieved by acting on the highest-ranked factors first, as they are the non-compensatory elements that theoretically unlock subsequent technology penetration and operational benefits. This strategic hierarchy was confirmed empirically, as the analysis of the other three healthcare organizations

**Table 9. The fuzzy scores and ranks of the factors for Healthcare C.**

| Factor | Score | Rank |
|--------|-------|------|
| OF1 | 0.0488 | 2 |
| OF2 | 0.0468 | 8 |
| OF3 | 0.0457 | 12 |
| OF4 | 0.0496 | 1 |
| OF5 | 0.0384 | 20 |
| TF1 | 0.0475 | 4 |
| TF2 | 0.0473 | 5 |
| TF3 | 0.0372 | 21 |
| TF4 | 0.0463 | 9 |
| TF5 | 0.0426 | 17 |
| TF6 | 0.0451 | 14 |
| EF1 | 0.0455 | 13 |
| EF2 | 0.0357 | 22 |
| EF3 | 0.0444 | 16 |
| EF4 | 0.0470 | 7 |
| MF1 | 0.0462 | 10 |
| MF2 | 0.0389 | 19 |
| MF3 | 0.0482 | 3 |
| MF4 | 0.0449 | 15 |
| IF1 | 0.0400 | 18 |
| IF2 | 0.0460 | 11 |
| IF3 | 0.0471 | 6 |

also revealed a broad alignment in the prioritization of key factors, with very slight variations in their relative rankings. This prioritization provides a clear mapping between the factors and their diffusion effect. The robustness of the results was also confirmed with sensitivity analysis in that the ranking of factors has not changed when Dombi parameter $\sigma$ was set at 2 and the prioritization is unaffected by small changes of aggregation parameters.

In terms of theoretical soundness and practical competence, the Fuzzy Delphi and Dombi LMAW method together offer a more flexible, accurate, and context-sensitive model of prioritizing complex factors than the majority of the current MCDM approaches. By using a two-stage methodological approach, we were able to treat the diffusion factors as important, non-compensatory features in a fuzzy decision system rather than just criteria. The Dombi LMAW supplied the accurate, context-sensitive feature weighting required to handle the uncertainty and interdependencies present in this high-complexity environment, while the FDM carried out the initial feature selection by eliminating non-consensus elements. By bringing the strategic planning process into line with accepted advanced fuzzy decision system concepts, this framework strengthens the theoretical rigor. The results are consistent with the wider theories of digital transformation and particularly in the developing countries where infrastructural and managerial preparedness can be the most valuable facilitators or hindrances of the diffusion of technology. Even though financial viability is an acknowledged issue, the comparatively mid-range score of Implementation Cost and the strikingly low score of Lack of ROI Clarity shows that economic issues are not seen to be particularly critical as opposed to more pressing operational issues. Also, the cross-ranking of Integration with IT infrastructure and Resistance to Change as the average-ranking factors, in their turn, highlights an overall focus on the technological integration and organizational flexibility of the diffusion of the blockchain in the health care supply chains. This is why there should be a top-down adoption of blockchain, where policy, infrastructure and leadership come first, and engagement at the individual level starts only when it is already established. The study

**Table 10. The fuzzy scores and ranks of the factors for Healthcare D.**

| Factor | Score | Rank |
|--------|-------|------|
| OF1 | 0.0487 | 2 |
| OF2 | 0.0462 | 10 |
| OF3 | 0.0458 | 12 |
| OF4 | 0.0505 | 1 |
| OF5 | 0.0387 | 20 |
| TF1 | 0.0476 | 5 |
| TF2 | 0.0478 | 4 |
| TF3 | 0.0385 | 21 |
| TF4 | 0.0463 | 9 |
| TF5 | 0.0436 | 17 |
| TF6 | 0.0454 | 14 |
| EF1 | 0.0457 | 13 |
| EF2 | 0.0382 | 22 |
| EF3 | 0.0446 | 16 |
| EF4 | 0.0472 | 6 |
| MF1 | 0.0464 | 8 |
| MF2 | 0.0389 | 19 |
| MF3 | 0.0482 | 3 |
| MF4 | 0.0450 | 15 |
| IF1 | 0.0424 | 18 |
| IF2 | 0.0461 | 11 |
| IF3 | 0.0466 | 7 |

combines rigor and practical relevance by adopting the hybrid technique, which combined the quantitative value of Dombi LMAW with the qualitative assessment procedure of Fuzzy Delphi. Policymakers, healthcare executives, and supply chain managers might benefit from the latter reflections' sector-specific and systematic suggestions as they work to implement blockchain technologies in resource-constrained environments. The research can offer a decision-support model that might be used to promote particular investments, legislative modifications, and capacity-building initiatives to facilitate the quicker adoption of blockchain in healthcare supply chains by reducing and prioritizing the most crucial elements.

## Theoretical Implications

This study's prioritization of Top Management Support (OF4) and Organizational Readiness (OF1) as the top two factors carries significant theoretical implications. While our MCDM approach determines the relative weight or importance of these factors, their consistent top ranking aligns with established diffusion and technology adoption theories, such as the Technology-Organization-Environment (TOE) framework [36]. According to these models, organizational elements like leadership support and preparedness serve as the cornerstone of any adoption process [35].

Additionally, the results empirically support the Organization component of the TOE framework as the main area of strategic focus in emerging economies, surpassing pressing issues with the Environmental (e.g., Innovation Pressure) and Technology (e.g., Smart Contract Automation) dimensions. This study makes a contribution to fuzzy set theory by showing how the Dombi operator produces a prioritization hierarchy that is more stable and theoretically justified than those generated by models assuming linear compensation among criteria by taking into account the high vagueness and non-linearity inherent in expert judgment. These elements serve as essential, non-compensatory threshold conditions for the successful diffusion of blockchain, especially in emerging economies [33]. In other words, a significant lack of leadership support

**Table 11. Rank of the factors for σ=2 for Healthcare A.**

| Factor | Fuzzy value | Score | Rank |
|--------|-------------|-------|------|
| OF1 | (4.46E-08, 1.43E-07, 4.5E-07) | 1.78E-07 | 2 |
| OF2 | (3.71E-08, 1.25E-07, 4.26E-07) | 1.61E-07 | 8 |
| OF3 | (3.42E-08, 1.18E-07, 4.17E-07) | 1.54E-07 | 12 |
| OF4 | (4.97E-08, 1.55E-07, 4.87E-07) | 1.93E-07 | 1 |
| OF5 | (8.3E-09, 7.06E-08, 3.1.0E-07) | 1.002E-07 | 20 |
| TF1 | (3.99E-08, 1.31E-07, 4.47E-07) | 1.69E-07 | 5 |
| TF2 | (4.03E-08, 1.32E-07, 4.57E-07) | 1.71E-07 | 4 |
| TF3 | (7.54E-09, 6.52E-08, 2.85E-07) | 9.221E-08 | 21 |
| TF4 | (3.68E-08, 1.24E-07, 4.16E-07) | 1.593E-07 | 9 |
| TF5 | (2.04E-08, 8.76E-08, 3.34E-07 | 1.176E-07 | 17 |
| TF6 | (3.34E-08, 1.17E-07, 4.14E-07) | 1.525E-07 | 14 |
| EF1 | (3.44E-08, 1.17E-07, 4.17E-07) | 1.534E-07 | 13 |
| EF2 | (6.41E-09, 6.07E-08, 2.84E-07) | 8.893E-08 | 22 |
| EF3 | (3.03E-08, 1.07E-07, 3.77E-07) | 1.396E-07 | 16 |
| EF4 | (3.96E-08, 1.3E-07, 4.42E-07) | 1.672E-07 | 6 |
| MF1 | (3.66E-08, 1.23E-07, 4.26E-07) | 1.582E-07 | 10 |
| MF2 | (8.75E-09, 7.5E-08, 3.2E-07) | 1.049E-07 | 19 |
| MF3 | (4.25E-08, 1.38E-07, 4.6E-07) | 1.756E-07 | 3 |
| MF4 | (3.26E-08,1.13E-07,3.93E-07) | 1.465E-07 | 15 |
| IF1 | (2.07E-08, 8.13E-08, 3.22E-07) | 1.114E-07 | 18 |
| IF2 | (3.6E-08, 1.22E-07, 4.25E-07) | 1.579E-07 | 11 |
| IF3 | (3.85E-08, 1.27E-07, 4.46E-07) | 1.658E-07 | 7 |

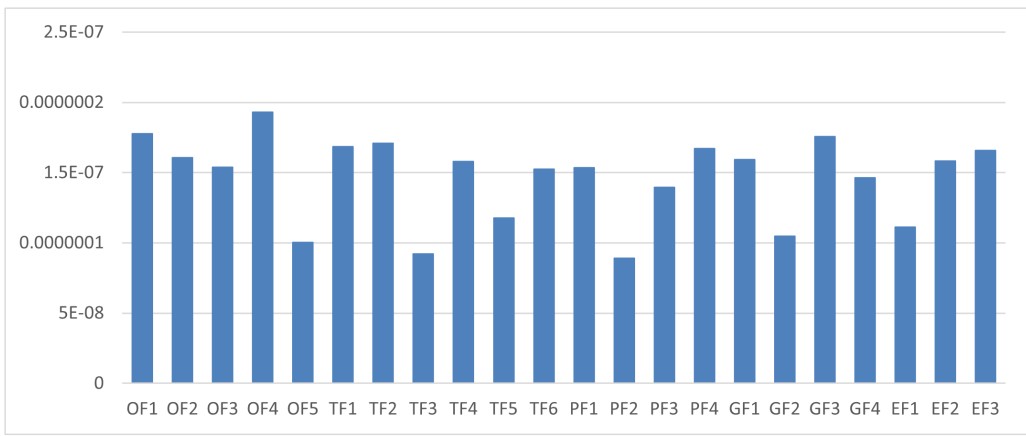

**Fig 5. Influence of σ on the ranking of factors for Healthcare A.** It can be seen that there was no change in the rank of the factors.

or fundamental readiness cannot be made up for by a high score in other factors (such as Innovation Pressure or Smart Contract Automation). While this study did not execute a formal Necessary Condition Analysis (NCA) to statistically test this causal attribute; the findings strongly suggest that OF4 and OF1 are necessary conditions. Future research should aim to formally test this hypothesis using NCA to enhance the theoretical persuasiveness of the prioritization.

## Managerial Implications

Blockchain technology represents an excellent opportunity for textile and healthcare manufacturers to advance their supply chains in terms of transparency and traceability. Manufacturers will get blockchain systems that will log every movement of raw materials, components, and finished goods, providing a clear visibility of the overall production process. Such transparency helps manufacturers track inefficiencies, make counterfeit goods less common and contribute to the best practices of supply chain management. Blockchain technology will help to streamline differences between healthcare suppliers and manufacturers in the procurement processes and improve collaboration among them. Instead of using the blockchain-based smart contracts to enable interactions with purchase agreements, payments and delivery schedules, the number of parties taking administrative tasks and disputes can be significantly reduced. Furthermore, financing solutions of blockchain-based supply chains can be offered to suppliers on affordable credit basis, thus making it easy for them to surmount the cash flow bottlenecks and post sustainable business growth [54]. Retailers within the healthcare sector can form trust and loyalty of customers by implementing blockchain technology in their services. Through enabling customers to see clear and verified information concerning the providence of items, the constituents and the manufacturing process, retailers are able to separate their brands and be confident to attract morally oriented customers. Also, transactions that are made via blockchain-based loyalty programs or rewards systems can encourage customers to make repeat purchases and hence create more bonds between the organizations and their customers [55]. Policymakers and industry body representatives are among the major drivers in the healthcare sector's blockchain adoption. Through the advancement of such industry-wide standards and guidelines for blockchain implementation, policymakers are able to ensure a refined legal framework that allows the promotion of new innovations and opportunities in collaboration. In addition, lawmakers should also explore how to use blockchain-based solutions for legal compliance, tax collection, and worker's rights protection, which will improve the sector's sustainability and integrity [56]. The blockchain technology in healthcare sector can bring not only revolutionary effects but also help increase supply chain transparency and leveraging the consumer trust. Integration of blockchain innovation is a key to value creation, strengthening of competitive positions and sustainable value for different revolves of industry.

## Conclusions

The study forms a novel and unique context-specific prioritizing framework through which assessment and prioritization of the key factors related to blockchain spread in the healthcare supply chain of Bangladesh are carried out. The incorporation of FDM and Dombi LMAW makes the research locally efficient since it achieves both methodological soundness and experts' agreement. Using the Fuzzy Delphi technique, 22 main sub-factors were confirmed out of the 26 factors that are contextually much less relevant, depending on consensus and defuzzified thresholds among the experts. This was then followed by use of the Dombi LMAW technique that was successful in ranking the factors as per the order of significance. The findings pinpoint at the systematic significance of such categories as Top Management Support, Organizational Readiness and Trust Among Stakeholders. Conversely, individual level concerns such as Innovation Pressure, Smart Contract Automation and Lack of ROI Clarity were determined as of relatively minor focus. The consistent prioritization of Top Management Support and Organizational Readiness across all case studies strongly suggests these factors act as necessary, non-compensatory threshold conditions for blockchain diffusion in resource-constrained environments, aligning with organizational elements of the TOE framework. The use of the Dombi LMAW enhanced the accuracy and robustness of this prioritization by effectively modeling the non-linear interdependencies among the factors. Regarding the applicable boundaries of the research, the five-dimensional factor framework (Organizational, Technological, Environmental, Managerial, and Individual) is theoretically generalizable, as it is based on established adoption models like the TOE framework [57]. However, the specific prioritization ranking found in this study is likely contingent on a specific set of contextual characteristics. The universal conditions for our conclusions can be determined through theoretical comparisons with other national contexts. For example, a theoretical comparison with India confirms that it shares similar boundary conditions to

Bangladesh, as its healthcare sector is also characterized by (a) significant resource constraints, including inadequate infrastructure and underfunding [58]; (b) a fragmented digital ecosystem still reliant on legacy systems with low interoperability [59]; and (c) regulatory ambiguity, not in government support for blockchain, but in the technology's conflict with new data privacy laws and the lack of a clear legal framework for Distributed Ledger Technology (DLT) applications beyond finance. These findings can be of practical use to decision-makers who want to create blockchain-powered solutions in healthcare logistics and operations.

### Limitations and future scopes

The study is based on relatively small sample of the expert opinions that can limit the applicability of the conclusions to other settings. Besides, even though 26 factors were identified within the context of the literature, there are other factors that might have been significant and which were not included because of contextual narrowing. Specifically, factors such as Vendor Support, Data Governance Frameworks and Digital Literacy were determined to be less immediate priorities by the expert consensus during the initial screening, reflecting the current low digital maturity level in the Bangladeshi healthcare sector where foundational organizational (OF) and managerial (MF) elements take precedence over advanced ecosystem and regulatory constructs [33,35,36]. This study strongly advocates for including these factors in future cross-country research to compare how blockchain diffusion priorities shift across varying national digital maturity levels.

Future work can also apply dynamic modeling methods (e.g., Fuzzy DANP or System Dynamics) to rigorously explore the evolving, time-dependent interdependencies among factors. Furthermore, future research should integrate parallel MCDM methods (such as AHP, BWM, or TOPSIS) to empirically compare rank stability and validate the superiority of the Dombi LMAW weighting in complex fuzzy environments, thereby enhancing the methodological discourse. The economic development level can be quantified using metrics such as Gross Domestic Product (GDP) per capita or the World Bank's income-group classification and the medical informatization level can be quantified using proxies such as national healthcare IT spending (as a percentage of GDP), the Digital Adoption Index (DAI) or EHR (Electronic Health Record) adoption rates within the country.

## Supporting information

**S1 File. Supporting tables.** This file contains S1 Table (Identified factors), S2 Table (Collected Data from the experts on identifying major factors), S3 Table (Defuzzification of the expert's opinion), and S4 Table (Data of the Factors Ratio Vectors for Healthcare A from the remaining five experts).
(DOCX)

## Author contributions

**Conceptualization:** Kazi Md. Tanvir Anzum.

**Data curation:** Nasif Morshed.

**Investigation:** Raad Shahamat Labib.

**Methodology:** Nasif Morshed, Raad Shahamat Labib, Mushfiqur Rahman.

**Software:** Raad Shahamat Labib.

**Supervision:** Kazi Md. Tanvir Anzum.

**Validation:** Nasif Morshed.

**Visualization:** Mushfiqur Rahman.

**Writing – original draft:** Nasif Morshed, Raad Shahamat Labib, Mushfiqur Rahman.

**Writing – review & editing:** Kazi Md. Tanvir Anzum.

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
