## [Decision Letter · Decision Letter 0]

16 Nov 2025

PONE-D-25-37578Strategic prioritization of blockchain diffusion factors by an integrated Fuzzy Delphi-Dombi LMAW model: A case study in Bangladesh Healthcare sectors.

PLOS ONE

Dear Dr. Tanvir,

Thank you for submitting your manuscript, **MS-PONE-D-25-37578**, titled *“Strategic prioritization of blockchain diffusion factors by an integrated Fuzzy Delphi-Dombi LMAW model: A case study in Bangladesh Healthcare sectors.”*

The reviewers have now evaluated your submission. After my assessment of their reports and the manuscript, the decision is **Major Revision**.

Your topic is relevant to healthcare technology adoption, and the methodological integration you propose has potential value. However, the reviewers identified several issues that must be resolved before the manuscript can be considered further. Addressing these points will significantly strengthen the rigor, transparency, and clarity of your study.

**Editorial Summary**

The most urgent concerns relate to:

insufficient methodological explanation of the Fuzzy Delphi thresholding and factor elimination processmissing ethics approval for expert involvementlimited expert sample sizeincomplete factor set justification and omission of relevant constructspresentation and language issues affecting overall clarity

Addressing these areas is essential for further consideration.

**Required Revisions**

Please respond to all reviewer comments in detail. In particular, you must:

**Provide ethics approval documentation** for the expert-involvement component or explain why it was exempt under your institutional policy. This is mandatory.**Strengthen methodological transparency** by fully explaining threshold determination in the Fuzzy Delphi stage and providing clear justification for factor elimination.**Expand or justify the expert sample size** and clarify selection criteria. If expansion is not possible, provide a rigorous explanation of sample sufficiency.**Reassess the factor set**, including Vendor Support, Data Governance, and Digital Literacy, and justify inclusion or exclusion explicitly.**Revise the manuscript for clarity and readability**, correcting language issues throughout and improving flow in the methods and results sections.

**Reviewer Reports**

For your convenience, the full reviewer comments are appended below.

**Reviewer 1**

The manuscript looks at the drivers and effect of the adoption of blockchain in Bangladesh’s healthcare supply chains, but I have a major concern with the presentation and method.

Comments:

1. The manuscript structure and language are affecting the reading flow. The language use at some point is very low (for example, a statement in the methods reads, "In order to fill in the details, we are expected to come up with the first step which is coming up with criteria from a detailed literature review and getting insights from the industry experts.").

2. Ethics committee clearance should be obtained for this kind of study to ensure adherence to ethical standards when involving expert opinions.

3. How are thresholds determined in the Fuzzy Delphi process, and explain properly how the factor elimination is justified to strengthen the credibility of factor selection?

**Reviewer 2**

1-The study relied on a small sample of only ten experts, which may have limited the generalizability of the results. Authors must expand the expert panel to include a larger and more diverse group of stakeholders to enhance the robustness and generalizability of the factor prioritization.

2-Some potentially relevant factors such as Vendor Support and Data Governance were omitted from the analysis due to the initial screening process. Authors must include additional factors such as Vendor Support, Data Governance, and Digital Literacy in subsequent iterations of the model to ensure comprehensiveness.

3-The research was conducted only in the Bangladeshi healthcare sector, which may restrict the transferability of findings to other sectors or countries. Authors should extend the research to other sectors or countries to enable cross-contextual comparisons and validate the model’s applicability.

4-The study used a static MCDM approach and did not account for the dynamic interdependencies between factors over time. Authors must adopt dynamic MCDM methods such as Fuzzy DANP or System Dynamics in future work to explore the evolving relationships between factors over time.

5- Authors must incorporate other MCDM techniques such as AHP, BWM, or TOPSIS in parallel to compare results and enhance the validity of the rankings.

6- The cross-sectional design did not capture longitudinal changes in factor importance or adoption behavior.

We look forward to receiving your revised manuscript.

Kind regards,

Asaad Ahmed Gad Elrab Ahmed

Academic Editor

PLOS ONE

Additional Editor Comments:

Reviewers' comments:

Reviewer's Responses to Questions

**Comments to the Author**

1. Is the manuscript technically sound, and do the data support the conclusions?

Reviewer #1: Partly

Reviewer #2: Yes

2. Has the statistical analysis been performed appropriately and rigorously? 

Reviewer #1: N/A

Reviewer #2: No

3. Have the authors made all data underlying the findings in their manuscript fully available?

Reviewer #1: No

Reviewer #2: Yes

4. Is the manuscript presented in an intelligible fashion and written in standard English?

Reviewer #1: No

Reviewer #2: Yes

5. Review Comments to the Author

Reviewer #1: The manuscript looks at the drivers and effect of the adoption of blockchain in Bangladesh’s healthcare supply chains, but I have a major concern with the presentation and method.

Comments:

1. The manuscript structure and language are affecting the reading flow. The language use at some point is very low (for example, a statement in the methods reads, "In order to fill in the details, we are expected to come up with the first step which is coming up with criteria from a detailed literature review and getting insights from the industry experts.").

2. Ethics committee clearance should be obtained for this kind of study to ensure adherence to ethical standards when involving expert opinions.

3. How are thresholds determined in the Fuzzy Delphi process, and explain properly how the factor elimination is justified to strengthen the credibility of factor selection?

Reviewer #2: 1-The study relied on a small sample of only ten experts, which may have limited the generalizability of the results. Authors must expand the expert panel to include a larger and more diverse group of stakeholders to enhance the robustness and generalizability of the factor prioritization.

2-Some potentially relevant factors such as Vendor Support and Data Governance were omitted from the analysis due to the initial screening process. Authors must include additional factors such as Vendor Support, Data Governance, and Digital Literacy in subsequent iterations of the model to ensure comprehensiveness.

3-The research was conducted only in the Bangladeshi healthcare sector, which may restrict the transferability of findings to other sectors or countries. Authors should extend the research to other sectors or countries to enable cross-contextual comparisons and validate the model’s applicability.

4-The study used a static MCDM approach and did not account for the dynamic interdependencies between factors over time. Authors must adopt dynamic MCDM methods such as Fuzzy DANP or System Dynamics in future work to explore the evolving relationships between factors over time.

5- Authors must incorporate other MCDM techniques such as AHP, BWM, or TOPSIS in parallel to compare results and enhance the validity of the rankings.

6- The cross-sectional design did not capture longitudinal changes in factor importance or adoption behavior.

6. PLOS authors have the option to publish the peer review history of their article (what does this mean?). If published, this will include your full peer review and any attached files.

Reviewer #1: No

Reviewer #2: No

---

## [Author Response · Author response to Decision Letter 1]

15 Dec 2025

Manuscript Number: PONE-D-25-37578

Manuscript Title: Strategic Prioritization of Blockchain Diffusion Factors by an Integrated Fuzzy Delphi–Dombi LMAW Model: A Case Study in Bangladesh Healthcare Sectors

Journal: PLOS ONE

Authors: Nasif Morsheda, Raad Shahamat Labiba, Mushfiqur Rahmana, Kazi Md. Tanvir Anzum

Corresponding Author: Kazi Md. Tanvir Anzum

Email: tanvir@iem.kuet.ac.bd

We sincerely thank the Editor and Reviewers for their constructive comments, which have significantly improved the quality, clarity, and rigor of our manuscript. We have carefully revised the manuscript to address all comments and suggestions. All changes in the revised manuscript are highlighted in red for ease of reference. Detailed responses are provided below.

Required Revision / Editorial Summary

Comment 1

Provide ethics approval documentation for the expert-involvement component or explain why it was exempt under your institutional policy. This is mandatory.

Authors’ Response:

We recognize that providing ethical oversight documentation or justification is a non-negotiable requirement. The study protocol was formally reviewed and deemed Exempt from full Institutional Review Board (IRB) review by the Office of the Director (Research & Extension), Khulna University of Engineering & Technology. Details of the exemption justification, expert recruitment period (April 1, 2024 – July 30, 2024), and confirmation of written informed consent from all participants have now been explicitly included.

Placement in Revised Manuscript:

Declaration section (highlighted in red).

Comment 2

Strengthen methodological transparency by fully explaining threshold determination in the Fuzzy Delphi stage and providing clear justification for factor elimination.

Authors’ Response:

We have strengthened the explanation of the Fuzzy Delphi screening process by explicitly reporting the consensus threshold value (S), calculated as the average of the defuzzified scores of the initial 26 factors. We have also clearly listed the four eliminated factors and justified their exclusion based on CVj values below the threshold.

Placement in Revised Manuscript:

Section 4.4.4 (Determination of Significant Factors) and Section 5 (Case Study), highlighted in red.

Comment 3

Expand or justify the expert sample size and clarify selection criteria. If expansion is not possible, provide a rigorous explanation of sample sufficiency.

Authors’ Response:

Expanding the expert panel was not feasible due to the reliance on highly specialized judgmental sampling. We have strengthened the justification by emphasizing the Delphi consensus principle and the intentional diversity of expert roles (clinical, managerial, technical, and policy) across government, private, and academic institutions to ensure construct validity.

Placement in Revised Manuscript:

Section 4.1 (Participants & Study Design), highlighted in red.

Comment 4

Reassess the factor set, including Vendor Support, Data Governance, and Digital Literacy, and justify inclusion or exclusion explicitly.

Authors’ Response:

We have explicitly justified the exclusion or low prioritization of these factors by noting the current focus on foundational adoption issues in the Bangladeshi healthcare context. These factors are now transparently discussed as limitations, with strong recommendations for inclusion in future cross-country and higher-maturity studies.

Placement in Revised Manuscript:

Section 8.1 (Limitations & Future Scopes), highlighted in red.

Comment 5

Revise the manuscript for clarity and readability, correcting language issues throughout and improving flow in the methods and results sections.

Authors’ Response:

A comprehensive language and grammar revision has been conducted across the entire manuscript. Sentence structure, academic tone, and logical flow—particularly in the Methods and Discussion sections—have been significantly improved to meet PLOS ONE standards.

Placement in Revised Manuscript:

Entire manuscript, highlighted in red.

Response to Reviewer #1

Comment 1

The manuscript structure and language affect reading flow, with instances of poor language quality.

Authors’ Response:

We have conducted a thorough language revision focusing on clarity, conciseness, and academic tone. The cited examples of poor sentence structure have been corrected, and overall readability has been improved.

Placement in Revised Manuscript:

Entire manuscript, highlighted in red.

Comment 2

Ethics committee clearance should be obtained for studies involving expert opinions.

Authors’ Response:

The study protocol was reviewed and granted IRB exemption by Khulna University of Engineering & Technology. Details regarding ethics exemption, consent procedures, and recruitment timeline are now clearly stated.

Placement in Revised Manuscript:

Declaration section, highlighted in red.

Comment 3

Clarify threshold determination and factor elimination in the Fuzzy Delphi process.

Authors’ Response:

We have explicitly reported the calculated threshold value (S) and identified the eliminated factors whose scores fell below this threshold, providing objective justification for the final factor set.

Placement in Revised Manuscript:

Section 4.4.4 and Section 5, highlighted in red.

Response to Reviewer #2

Comment 1

The small expert sample size may limit generalizability.

Authors’ Response:

We have strengthened the justification of the n = 10 expert panel based on Delphi consensus methodology and intentional role diversity, which is appropriate for MCDM-based expert studies.

Placement in Revised Manuscript:

Section 4.1 (Participants & Study Design), highlighted in red.

Comment 2

Potentially relevant factors were omitted due to initial screening.

Authors’ Response:

The exclusion of these factors is now clearly justified based on contextual priorities, and they are explicitly acknowledged as limitations and future research directions.

Placement in Revised Manuscript:

Section 8.1 (Limitations & Future Scopes), highlighted in red.

Comment 3

The single-country focus may restrict transferability.

Authors’ Response:

We have added a discussion on contextual boundaries and emphasized the need for future cross-sectoral and cross-country validation.

Placement in Revised Manuscript:

Section 8 (Conclusion), highlighted in red.

Comment 4

The static MCDM approach does not capture dynamic interdependencies.

Authors’ Response:

We now explicitly acknowledge this limitation and recommend future adoption of dynamic methods such as Fuzzy DANP or System Dynamics.

Placement in Revised Manuscript:

Section 8.1 (Limitations & Future Scopes), highlighted in red.

Comment 5

Parallel MCDM methods should be incorporated for validation.

Authors’ Response:

We have added a recommendation for future studies to integrate alternative MCDM techniques (AHP, BWM, TOPSIS) for comparative validation.

Placement in Revised Manuscript:

Section 8.1 (Limitations & Future Scopes), highlighted in red.

Comment 6

The cross-sectional design does not capture longitudinal changes.

Authors’ Response:

We have clarified the cross-sectional nature of the study and linked this limitation to the need for future longitudinal and dynamic analyses.

Placement in Revised Manuscript:

Section 4.1 and Section 8.1, highlighted in red.

We believe that the revised manuscript has fully addressed all editorial and reviewer comments and now meets the standards of PLOS ONE. We sincerely appreciate the opportunity for revision and look forward to your positive reassessment.

---

## [Editor Report · Decision Letter 1]

23 Dec 2025

PONE-D-25-37578R1Strategic prioritization of blockchain diffusion factors by an integrated Fuzzy Delphi-Dombi LMAW model: A case study in Bangladesh Healthcare sectors.PLOS One

Dear Dr. Tanvir,

Thank you for submitting your manuscript to PLOS ONE. After careful consideration, we feel that it has merit but does not fully meet PLOS ONE’s publication criteria as it currently stands. Therefore, we invite you to submit a revised version of the manuscript that addresses the points raised during the review process.

he reviewers have now evaluated your submission. After my assessment of their reports and the manuscript, the decision is Major Revision.

Your topic is relevant to healthcare technology adoption, and the methodological integration you propose has potential value. However, the reviewers identified several issues that must be resolved before the manuscript can be considered further. Addressing these points will significantly strengthen the rigor, transparency, and clarity of your study.

Editorial Summary

The most urgent concerns relate to:

•     insufficient methodological explanation of the Fuzzy Delphi thresholding and factor elimination process

•     missing ethics approval for expert involvement

•     limited expert sample size

•     incomplete factor set justification and omission of relevant constructs

•     presentation and language issues affecting overall clarity

Addressing these areas is essential for further consideration.

Required Revisions

Please respond to all reviewer comments in detail. In particular, you must:

1.     Provide ethics approval documentation for the expert-involvement component or explain why it was exempt under your institutional policy. This is mandatory.

2.     Strengthen methodological transparency by fully explaining threshold determination in the Fuzzy Delphi stage and providing clear justification for factor elimination.

3.     Expand or justify the expert sample size and clarify selection criteria. If expansion is not possible, provide a rigorous explanation of sample sufficiency.

4.     Reassess the factor set, including Vendor Support, Data Governance, and Digital Literacy, and justify inclusion or exclusion explicitly.

5.     Revise the manuscript for clarity and readability, correcting language issues throughout and improving flow in the methods and results sections.

Reviewer Reports

For your convenience, the full reviewer comments are appended below.

We look forward to receiving your revised manuscript.

Kind regards,

Asaad Ahmed Gad Elrab Ahmed

Academic Editor

PLOS One
---

## [Author Response · Author response to Decision Letter 2]

27 Jan 2026

MANUSCRIPT INFORMATION

Manuscript Number: PONE-D-25-37578

Manuscript Title: Strategic prioritization of blockchain diffusion factors by an integrated Fuzzy Delphi-Dombi LMAW model: A case study in Bangladesh healthcare sectors.

Journal: PLOS ONE

We appreciate the constructive feedback provided by the editors and reviewers of PLOS ONE, which helped to make the manuscript better. The authors have prepared a revised manuscript that takes into account all of the reviewers' and editor comments, suggestions and recommendations. Some elements of the manuscript's presentation and orientation have been altered in response to the reviewers' recommendations. For ease of reference, manuscript changes have been highlighted in red text. Below is a description of the authors' responses to the editors’ and reviewers' remarks:

Reviewer Comments and Authors’ Responses

Reviewer Comment 1

Provide ethics approval documentation for the expert-involvement component or explain why it was exempt under your institutional policy. This is mandatory.

Authors’ Response:

We acknowledge that ethical oversight documentation or a formal exemption statement is a mandatory requirement. Accordingly, the study protocol was formally reviewed and classified as Exempt from full Institutional Review Board (IRB) review by the Office of the Director (Research & Extension), Khulna University of Engineering & Technology (KUET). A detailed justification for this exemption, confirmation of the expert recruitment period (April 1, 2024 to July 30, 2024), and documentation of informed written consent obtained from all participants have now been explicitly incorporated into the manuscript. These additions confirm full compliance with ethical research standards.

Placement in Revised Manuscript:

Participants and Study Design (Line 248-256), highlighted in red text.

Reviewer Comment 2

Strengthen methodological transparency by fully explaining threshold determination in the Fuzzy Delphi stage and providing clear justification for factor elimination.

Authors’ Response:

To enhance methodological transparency, we have expanded the explanation of the threshold determination process in the Fuzzy Delphi stage. Specifically, the consensus threshold (S = 6.9) is now clearly defined as the arithmetic mean of the defuzzified scores of the initial 26 candidate factors. We further clarified that this threshold functions as a screening mechanism to retain only those factors demonstrating strong expert consensus for inclusion in the subsequent Dombi LMAW prioritization stage, thereby improving the robustness and precision of the final results.

Additionally, we strengthened the justification for factor elimination by providing a contextual interpretation alongside the numerical criterion. The exclusion of four factors (OF6, TF7, EF5, and IF4) is now explained as reflecting the expert panel’s prioritization of foundational organizational and leadership-related factors over advanced technical or individual-level constructs, which aligns with the current digital maturity level of the Bangladeshi healthcare sector.

Placement in Revised Manuscript:

Determination of Significant Factors (Lines 318–326) and Case Study section (Lines 413–421), highlighted in red text.

Reviewer Comment 3

Expand or justify the expert sample size and clarify selection criteria. If expansion is not possible, provide a rigorous explanation of sample sufficiency.

Authors’ Response:

We have provided a rigorous methodological justification for the expert sample size (n = 10) by grounding it in the consensus-based principles of the Delphi method. The revised manuscript now includes a supporting citation (Hsu & Sandford, n.d.), which establishes that a panel of 10–15 experts is sufficient to achieve stable and reliable consensus in Delphi-based studies. Furthermore, we clarified that purposive sampling was employed to ensure balanced representation across four critical functional domains—clinical, managerial, technical, and policy-making—thereby strengthening construct validity and ensuring comprehensive domain expertise.

Placement in Revised Manuscript:

Participants and Study Design section (Lines 229–245), highlighted in red text.

Reviewer Comment 4

Reassess the factor set, including Vendor Support, Data Governance, and Digital Literacy, and justify inclusion or exclusion explicitly.

Authors’ Response:

We acknowledge the relevance and importance of Vendor Support, Data Governance, and Digital Literacy in blockchain diffusion. In response, we have explicitly clarified their status in the revised manuscript. The exclusion or lower prioritization of these factors is now justified by the expert panel’s assessment that the current healthcare context in Bangladesh is primarily focused on foundational organizational readiness and managerial commitment rather than advanced ecosystem-level or implementation-stage considerations. To ensure transparency, these factors are explicitly discussed in the Limitations section and identified as promising avenues for future research, particularly in cross-country or higher digital-maturity comparative studies.

Placement in Revised Manuscript:

Limitations and Future Research Directions section (Lines 678–685), highlighted in red text.

Reviewer Comment 5

Revise the manuscript for clarity and readability, correcting language issues throughout and improving flow in the methods and results sections.

Authors’ Response:

We have undertaken a comprehensive and systematic revision of the manuscript to improve clarity, coherence, and academic readability. This includes thorough language editing across all sections, with particular emphasis on the Methods, Results, and Discussion sections. All previously identified issues related to grammar, sentence structure, and flow have been addressed to ensure alignment with the stylistic and scholarly standards of PLOS ONE.

Placement in Revised Manuscript:

Entire manuscript, with all revisions highlighted in red text.

Therefore, the author concluded that, the revised manuscript has addressed all comments of the reviewers and editors. The orientation and presentation of the paper were changed in the revised manuscript to address all reviewers’ and editors’ comments. All changes are highlighted in red text in the revised manuscript. Hence, we look forward to your positive reassessment.

---

## [Decision Letter · Decision Letter 2]

20 Apr 2026

Strategic prioritization of blockchain diffusion factors by an integrated Fuzzy Delphi-Dombi LMAW model: A case study in Bangladesh Healthcare sectors.

PONE-D-25-37578R2

Dear Dr. Kazi Md. Tanvir Anzum Tanvir,

We’re pleased to inform you that your manuscript has been judged scientifically suitable for publication and will be formally accepted for publication once it meets all outstanding technical requirements.

Before this, you are required to revise the last part of the abstract (lines 28-33) by deleting the study limitaions and instead, state the implications of the study.

Kind regards,

Sulemana Bankuoru Egala (PhD)

Academic Editor

PLOS One

Additional Editor Comments (optional):

Reviewers' comments:

Reviewer's Responses to Questions

**Comments to the Author**

1. If the authors have adequately addressed your comments raised in a previous round of review and you feel that this manuscript is now acceptable for publication, you may indicate that here to bypass the “Comments to the Author” section, enter your conflict of interest statement in the “Confidential to Editor” section, and submit your "Accept" recommendation.

Reviewer #1: All comments have been addressed

Reviewer #3: All comments have been addressed

2. Is the manuscript technically sound, and do the data support the conclusions?

Reviewer #1: Yes

Reviewer #3: Yes

3. Has the statistical analysis been performed appropriately and rigorously? 

Reviewer #1: Yes

Reviewer #3: Yes

4. Have the authors made all data underlying the findings in their manuscript fully available?

Reviewer #1: No

Reviewer #3: Yes

5. Is the manuscript presented in an intelligible fashion and written in standard English?

Reviewer #1: Yes

Reviewer #3: Yes

6. Review Comments to the Author

Reviewer #1: The authors addressed all comments from the last round of review, which has improved the quality and structure of the manuscript.

One minor concern is the addition of the limitation and future direction in the abstract, which has been addressed in lines 675-693. It should be removed to allow the abstract to flow properly.

Reviewer #3: The second revision has been well accommodated. The author (s) addressed all the aspects. It is publishable now.

7. PLOS authors have the option to publish the peer review history of their article (what does this mean?). If published, this will include your full peer review and any attached files.

Reviewer #1: No

Reviewer #3: **Yes:** Dr. SYED FAR ABID HOSSAIN

---

## [Editor Report · Acceptance letter]

PONE-D-25-37578R2

PLOS One

Dear Dr. Tanvir Anzum,

I'm pleased to inform you that your manuscript has been deemed suitable for publication in PLOS One. Congratulations! Your manuscript is now being handed over to our production team.

Kind regards,

on behalf of

Prof. Sulemana Bankuoru Egala

Academic Editor

PLOS One